# Acetyl-carnitine improves hyperactivity and learning deficits in *KAT6A* haploinsufficient mice

Samantha Eccles[1,2] , Hannah K Vanyai[1,2], Maria I Bergamasco[1,2], Shezlie Malelang[1,2] , Havva Pehlivanoglu[1], Alexandra L Garnham[1,2] , Nishika Ranathunga[1] , Marnie E Blewitt[1,2], Adam P Vogel[3,4] , Gordon K Smyth[1,5] , Anthony J Hannan[6,7], Tim Thomas[1,2,*] , Anne K Voss[1,2,*]

**Pathogenic variants in one allele of the *KAT6A* gene encoding the histone acetyltransferase KAT6A (MOZ, MYST3) cause Arboleda–Tham syndrome (ARTHS), characterised by developmental delay, cognitive impairment, and autism-like behaviours. As histone acetylation is reversible, and brain development continues after birth, treatments that address deficits in histone acetylation may ameliorate the condition. Here, we examined the effects of ARTHS mutations on histone acetylation in human cells and the effects of heterozygous loss of *Kat6a* in mice (*Kat6a*[+/−]) on learning, memory, activity, and sociability. We found that KAT6A was required for normal levels of histone H3 lysine 23 acetylation (H3K23ac) in human cells and mouse brain. *Kat6a*[+/−] mice displayed hyperactivity and learning, memory, and sociability deficits compared with WT mice. Treatment with the acetyl-donor, acetyl-L-carnitine (ALCAR) resulted in the rescue of H3K23ac levels in mouse brain and amelioration of the hyperactivity and learning impairments. Our results suggest that some individuals with ARTHS might benefit from ALCAR treatment. However, the suitability of ALCAR treatment would depend on the specific *KAT6A* variant and should be discussed with health professionals.**

## Introduction

Mutations of one allele of the gene encoding the histone acetyltransferase KAT6A (monocytic leukaemia zinc finger protein [MOZ], MYST family member 3 [MYST3]) cause the autosomal dominant cognitive disorder, Arboleda–Tham syndrome (ARTHS; OMIM # 616268 [Arboleda et al, 2015; Tham et al, 2015]). Individuals with ARTHS present with developmental delay, cognitive impairment (100% penetrance), delay or absence of speech (100%), facial dysmorphism (85%), feeding difficulties (78%), and neonatal hypotonia (76%) in a study of 76 individuals with *KAT6A* mutations (Kennedy et al, 2019). Other anomalies observed include visual defects (63%), cardiac defects (51%), constipation (51%), frequent infections (47%), behavioural problems (39%), sleep disturbances (37%), and microcephaly (31%) (Kennedy et al, 2019). Autism-like behaviours (33%) were reported in a study of 49 individuals with *KAT6A* mutations (St John et al, 2022). Hyperactivity (28%) was identified in a study of 14 individuals with ARTHS and autism-like behaviours (28%) confirmed, including reduced social awareness, social cognition, and social communication, as well as restricted interests and repetitive behaviours (Ng et al, 2024).

KAT6A is a member of the MYST family of histone acetyltransferases, which include the closely related KAT6B, as well as KAT5, KAT7, and KAT8 (Voss & Thomas, 2018). KAT6A is a large 2004 aa protein containing multiple functional protein domains (Fig 1A). Via its MYST family histone lysine acetyltransferase (HAT) domain, KAT6A has been proposed to acetylate histone H3 at lysine 14 (H3K14) in HEK293T cells (Doyon et al, 2006), H3K9 in mouse embryos (Voss et al, 2009, 2012), and H3K23 in glioblastoma cells (Lv et al, 2017). Via its double plant homeodomain (PHD) finger, KAT6A has been proposed to bind unmodified H3R2 and acetylated or crotonylated H3K14 (Qiu et al, 2012; Dreveny et al, 2014; Sharma et al, 2023). KAT6A has been shown to bind DNA at unmethylated CpG islands via the first of its two amino-terminal winged helix domains (WHD [Becht et al, 2023; Weber et al, 2023]) and to interact with the runt family transcription factor RUNX2 via its carboxy-terminal methionine- and serine-rich regions (Pelletier et al, 2002). KAT6A occurs in a protein complex with the chromatin adaptor proteins BRPF1, BRPF2, or BRPF3, ING4 or ING5, and MEAF6 (Doyon et al, 2006). BRPF1 contains a bromodomain, which is reported to bind H2AK5ac, H4K12ac, and H3K14ac (Poplawski et al, 2014), and two PHD fingers closely linked by a zinc knuckle, which bind to unmodified histone H3 tail and nucleosomal DNA (Klein et al, 2016).

[1]The Walter and Eliza Hall Institute of Medical Research, Melbourne, Australia    [2]Department of Medical Biology, University of Melbourne, Melbourne, Australia    [3]School of Health Sciences, University of Melbourne, Melbourne, Australia    [4]Redenlab Ltd., Melbourne, Australia    [5]School of Mathematics and Statistics, University of Melbourne, Parkville, Australia    [6]Florey Institute of Neuroscience and Mental Health, Melbourne Brain Centre, University of Melbourne, Parkville, Australia    [7]Department of Anatomy and Neuroscience, University of Melbourne, Parkville, Australia

Correspondence: tthomas@wehi.edu.au; avoss@wehi.edu.au
*Tim Thomas and Anne K Voss share equal senior authorship

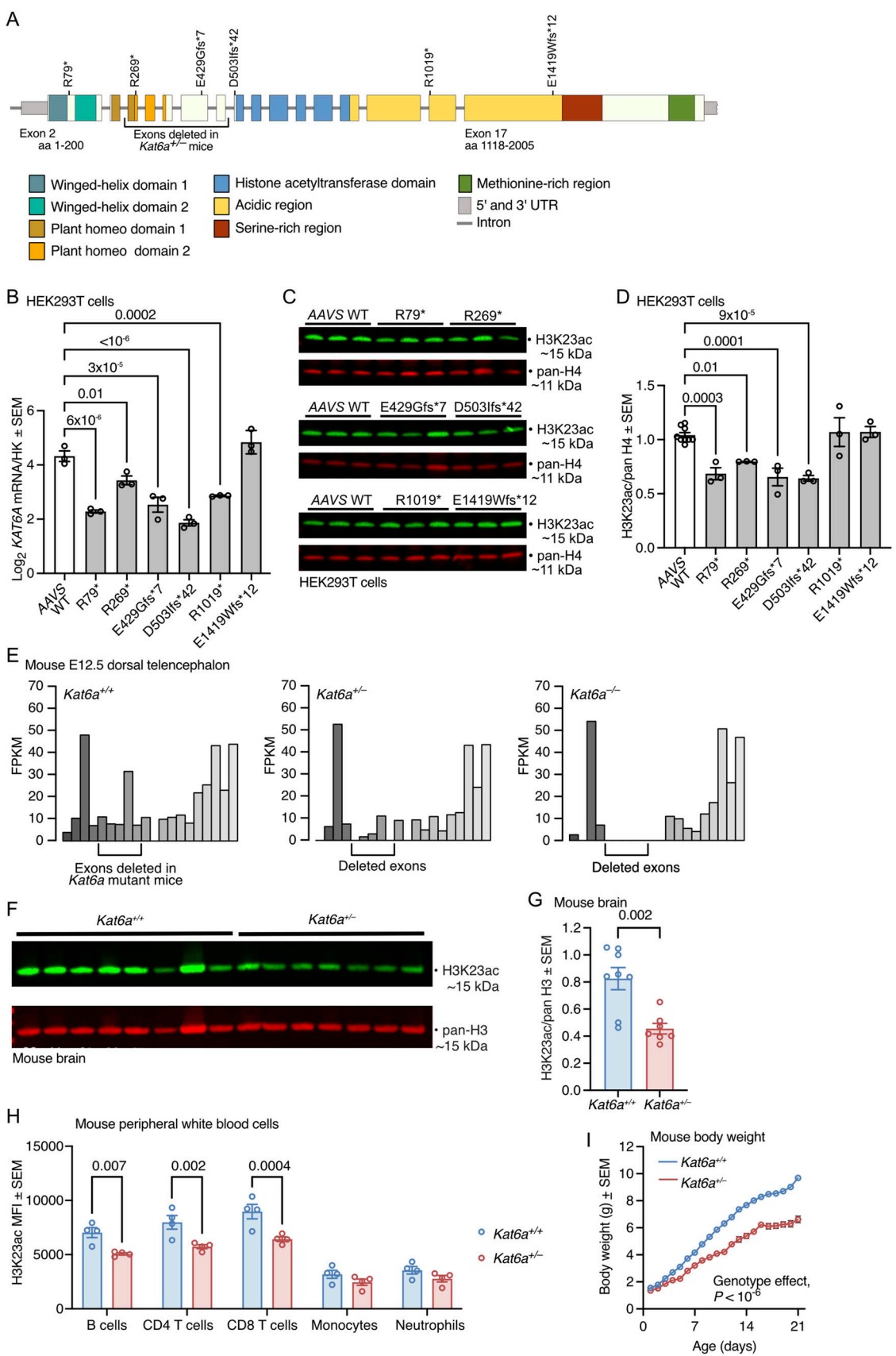

The PHD finger of ING4 and ING5 binds to methylated H3K4 (Champagne et al, 2008; Palacios et al, 2010; Ormaza et al, 2019). Together with the DNA (WHD) and histone-binding capability (PHD) of KAT6A itself, these chromatin adaptor proteins ensure secure recruitment of KAT6A to the chromatin, where the HAT domain can access and acetylate its histone lysine target(s). Because KAT6A contains multiple functional domains, it is possible that ARTHS variants may result in a truncated protein that retains acetyl-transferase activity but lacks protein–protein interaction domains that regulate this activity.

ARTHS mutations occur along the entire length of the KAT6A protein within all defined protein domains (WHD, PHD, HAT), as well as the less defined large acidic region (aa 780 to 1,438) and the carboxy-terminal methionine- and serine-rich regions (Kennedy et al, 2019). The collection of KAT6A variants associated with ARTHS also includes very early truncations, which are likely to represent complete loss of function of one allele of KAT6A, for example, loss of the start codon in c.1A>G, p.M1V or early truncations including at amino acid 65 of 2004 in c.195_198del, p.Asn65Lysfs*15 (Kennedy et al, 2019).

Chromatin proteins are critical for prenatal development. Heterozygous disruption of genes encoding chromatin-associated proteins commonly causes cognitive impairment syndromes in which other organ systems can also be affected (Fahrner & Bjornsson, 2014; Iwase et al, 2017). Homozygous null deletion of the Kat6a gene in mice causes foetal to perinatal death (depending on the genetic background), characterised by cleft palate, shortened jaws, cardiac septum defects, and interrupted aortic arch, as well as an anterior homeotic trans-formation of 19 body segments affecting the vertebral column and the nervous system (Voss et al, 2009, 2012; Vanyai et al, 2015). Congruently, the loss of KAT6A results in a shift in the expression of HOX genes (body segment identity), and the reduction in the expression of DLX genes (jaw and palate development) and TBX genes (heart development), among other affected genes (Voss et al, 2009, 2012; Vanyai et al, 2015). The loss of one allele of Kat6a causes isolated cardiac septum defects with 10% penetrance (Voss et al, 2012).

Histone acetylation is a dynamic process catalysed by lysine acetyltransferases and erased by histone deacetylases. Therefore, in principle any changes are reversible, and indeed, histone acetylation levels can be decreased and increased, respectively, by inhibiting acetyltransferases (Baell et al, 2018; Sharma et al, 2023; Mukohara et al, 2024) or deacetylases (Alarcón et al, 2004; Bjornsson et al, 2014; Jung et al, 2017; Bergamasco et al, 2024). Furthermore, providing more acetyl-groups by treatment with an acetyl-donor can increase histone acetylation levels (Bergamasco et al, 2024). Brain development continues after birth (Stiles & Jernigan, 2010; Semple et al, 2013) with some processes peaking between after 3 yr of age (Huttenlocher & Dabholkar, 1997). For example, mean synaptic density peaks at 8 mo of age in the visual cortex and at 3.6 yr of age in the auditory and prefrontal cortex, after which synapse elimination eventually results in adult levels of synaptic density during late adolescence (Huttenlocher & Dabholkar, 1997). Grey matter volume peaks at 12 yr of age (Semple et al, 2013). Of note, histological and cell biological changes in the developing brain are accompanied by functional changes. For example, executive functions including logical rea-soning and judgement appear to develop gradually during childhood and adolescence (Sternberg, 1979; Sternberg & Rifkin, 1979). Therefore, an opportunity may present itself to ameliorate ARTHS by treatment increasing histone acetylation in individuals with a reduction in histone acetylation. Furthermore, Kat6a is expressed in the adult brain (Kraft et al, 2011; Dillman et al, 2013), and thus, there might be an ongoing requirement for KAT6A for brain function, which might also benefit from treatment increasing histone acetylation levels.

Thus, we set out to determine, first, whether $Kat6a^{+/-}$ mice could be used as a model for ARTHS with respect to effects on histone acetylation and cognitive and social functions and, second, whether treatment aimed to increase histone acetylation levels might ameliorate any learning, memory, or social dysfunction that might be present.

# Results

### KAT6A mutations cause a decrease in histone H3 lysine 23 (H3K23) acetylation in human cells and mice

To determine the effects of mutation of the KAT6A gene on histone acetylation, we generated human HEK293T cells carrying

---

**Figure 1. KAT6A mutations causing Arboleda–Tham syndrome (ARTHS) and heterozygous deletion of Kat6a in mice cause a decrease in histone H3 lysine 23 (H3K23) acetylation.**

**(A)** Schematic drawing of the exon–intron structure of the KAT6A gene with the protein domains encoded, the ARTHS mutations examined, and the exons deleted in mice indicated (exon numbering as in NM_001364449.1). **(B, C, D)** KAT6A mRNA levels normalised to housekeeping genes (HK) GAPDH and HSP90AB1 assessed by RT–qPCR, (B) and H3K23 acetylation levels and pan-histone H3 levels assessed by Western blotting (C) and densitometry (D) in HEK293T cells with ARTHS-causing mutations introduced by CRISPR/Cas9 homology-directed gene editing. 0.5 μg of acid-extracted protein was loaded per lane (C). **(E)** Representative sample of RNA-sequencing fragment counts collected over individual Kat6a exons of dorsal telencephalon isolated from $Kat6a^{+/+}$ WT, $Kat6a^{+/-}$ heterozygous, and $Kat6a^{-/-}$ homozygous E12.5 mouse embryos. The position of the deleted exons 5–9 is indicated (exon numbering as in NM_001081149.2). **(F, G)** H3K23 acetylation levels and pan-histone H3 levels assessed by Western blotting (F) and densitometry (G) in $Kat6a^{+/+}$ WT and $Kat6a^{+/-}$ heterozygous adult mouse brain. 1 μg of acid-extracted protein was loaded per lane (F). **(H)** Intranuclear flow cytometry analysis of median fluorescence intensity (MFI) of H3K23ac minus isotype control in $Kat6a^{+/+}$ WT and $Kat6a^{+/-}$ heterozygous adult mouse peripheral white blood cells. **(I)** Daily body weight development of $Kat6a^{+/-}$ versus $Kat6a^{+/+}$ mice over the first 21 d of life. Effects of genotype, $P < 10^{-6}$. N = 3 independently genetically modified cell clones per ARTHS-causing mutation (B, D); and 4 (E), 7–8 (G), 4 (H), and 21 mice per genotype (I). Each circle in represents an independently modified cell clone (B, D) or an individual mouse (G, H). Each Western blot lane was loaded with protein from an independently modified cell clone in (C) and an individual mouse (F). Data are displayed as the mean ± SEM (B, D, G, I), RNA-sequencing fragments (reads) per kilobase per one million fragments in the library (FPKM; (E)), and median ± SEM (H). Data were analysed by one-way ANOVA with Dunnett's multiple comparison test (B, D), unpaired, two-tailed t test (G), two-way ANOVA with Šidák's multiple comparison test (H, I), and as described in detail under RNA-sequencing data analysis in the Materials and Methods section (E). Source data are available for this figure.

six different *KAT6A* variants observed in individuals with ARTHS using CRISPR/Cas9 and homology-directed repair (Tables S1, S2, and S3). The mutations chosen (c.235C>T, p.R79*; c.805C>T, p.R269*; c.1283_1284insT, p.E429Gfs*7; c.1507delG, p.D503Ifs*42; c.3055C>T, p.R1019*; c.4254_4257del (tgag), p.E1419Wfs*12) spanned the length of the KAT6A-coding region (Fig 1A). Analysis of clonal HEK293T cell lines with these ARTHS mutations revealed a reduction in *KAT6A* mRNA ($P < 10^{-6}$ to $P = 0.01$) in cell lines with all mutations except the most carboxy-terminal mutation, E1419Wfs*12 (Fig 1B), consistent with nonsense-mediated RNA decay and protection against decay of RNA with mutations in the final coding exon. Western blotting showed that the first four more amino-terminal mutations ($P = 9 \times 10^{-5}$ to 0.01; R79*, R269*, E429Gfs*7, D503Ifs*42; Fig 1C and D) caused a reduction in histone H3 lysine 23 acetylation (H3K23ac), whereas the last two more carboxy-terminal mutations (R1019*, E1419Wfs*12) did not affect H3K23a levels, suggesting that these two mutations might result in residual production of the KAT6A protein with histone acetyltransferase activity. Because of the lack of good-quality anti-KAT6A antibodies, we were unable to determine KAT6A protein levels. The acetylation levels of H3K9 and H3K14 were slightly reduced in HEK293T cell clones with only one of the six ARTHS mutations, each (Fig S1A–D), suggesting that H3K23 was the major histone acetylation mark affected by mutation of one allele of *KAT6A*. The reduction of *KAT6A* mRNA levels in HEK293T cell clones with five of the six ARTHS mutations (Fig 1B) suggested that mice lacking one allele of *Kat6a* would be a suitable model for a subset of individuals with ARTHS and would allow examination of acetylation levels in the brain, neuronal functions, cognition, behaviour, and gene expression in brain tissue.

Mouse telencephalon lacking both alleles of *Kat6a* (*Kat6a*$^{-/-}$ [Voss et al, 2009]) completely lacked RNA-sequencing reads mapping to the deleted *Kat6a* exons 5–9 (Fig 1E). Lack of one allele of *Kat6a* (*Kat6a*$^{+/-}$) resulted in a reduction of RNA reads over *Kat6a* exons 5–9 (Fig 1E) and a reduction in H3K23ac in the adult brain ($P = 0.002$; Fig 1F and G) and in peripheral B and T cells ($P = 0.0004–0.007$; Fig 1H). These results suggested that with respect to *Kat6a* mRNA levels and H3K23ac levels, *Kat6a*$^{+/-}$ mice behaved similar to HEK293T cell clones with five and four, respectively, of the six ARTHS mutations (Fig 1B–D).

### *Kat6a*$^{+/-}$ mice are small with normal brain anatomy and histology

*Kat6a*$^{+/-}$ mice had a normal body weight at birth but displayed a lower weight gain over the first 3 wk of life compared with WT littermate controls ($P < 10^{-6}$; Fig 1I) and a smaller body size (Fig S2A). *Kat6a*$^{+/-}$ mice remained small throughout adulthood ($P < 10^{-6}$; Fig S2B). The brain weight corrected for body weight was similar in *Kat6a*$^{+/-}$ and *Kat6a*$^{+/+}$ mice (Fig S2C). The whole brain and the cerebral cortex, the cortical layers, and the ventricles of *Kat6a*$^{+/-}$ and *Kat6a*$^{+/+}$ mice displayed a similar morphology and morphometry (Fig S2D–L). Cortical neurons isolated from 4-wk-old *Kat6a*$^{+/-}$ and *Kat6a*$^{+/+}$ mice and cultured in vitro displayed similar numbers of neurons, neurites, neurite branching, and neurite length (Fig S3A–D). Overall, *Kat6a*$^{+/-}$ mice did not appear to have major structural anomalies of the brain.

### *Kat6a* haploinsufficiency affects the expression of genes required for brain development and function

Histone acetylation is generally associated with active gene transcription (Wang et al, 2009; Stasevich et al, 2014). To assess the effects of loss of one allele of *Kat6a* on mRNA levels in the cortical neurons, we conducted RNA sequencing on E16.5 cortical neurons. Because *Kat6a*$^{-/-}$ foetuses die between E13.5 and E14.5 on a *C57BL/6* genetic background, we had only *Kat6a* WT and heterozygous samples available at E16.5. The loss of one allele of most genes does not result in obvious defects, and the effects of loss of one allele of *Kat6a* on gene expression were mild. Multidimensional scaling showed that the expression profiles of *Kat6a*$^{+/-}$ and *Kat6a*$^{+/+}$ E16.5 cortical neuron samples segregated between genotypes in dimension 1 (Fig S4A). In the E16.5 cortical neurons, 16,714 genes were expressed at sufficient levels to warrant differential gene expression analysis. Of these, only 20 were differentially expressed in *Kat6a*$^{+/-}$ versus *Kat6a*$^{+/+}$ cortical neurons, and only if an FDR of less than 0.1 was applied, 7 genes were down-regulated and 13 genes were up-regulated (Fig S4B and C; Table S4). Although loss of one allele of *KAT6A* has strong effects on cognition in humans and *Kat6a*$^{+/-}$ versus *Kat6a*$^{+/+}$ mouse cortical neuron samples segregated by genotype (Fig S4A), the differential expression analysis was not sensitive enough to detect major effects on gene expression in E16.5 cortical neurons.

To enhance our ability to detect differential gene expression, we conducted RNA sequencing on E12.5 dorsal telencephalon, at which developmental stage we had *Kat6a* WT, heterozygous, and homozygous null samples available. *Kat6a*$^{-/-}$ and *Kat6a*$^{+/+}$ E12.5 dorsal telencephalon samples segregated primarily in dimension 1 and to a lesser degree in dimension 2 (Fig 2A). *Kat6a*$^{+/-}$ samples were positioned intermediate between *Kat6a*$^{-/-}$ and *Kat6a*$^{+/+}$ samples (Fig 2A). Of 17,043 expressed genes, 70 were down-regulated and 70 up-regulated in *Kat6a*$^{-/-}$ versus *Kat6a*$^{+/+}$ E12.5 dorsal telencephalon (FDR < 0.05; Fig 2B; Table S5 (a)). In *Kat6a*$^{+/-}$ versus *Kat6a*$^{+/+}$ samples, 64 genes were down-regulated and 256 up-regulated (FDR < 0.5; Fig S4D; Table S6 (a)). The *Kat6a*$^{+/-}$ samples were positioned intermediate between *Kat6a*$^{-/-}$ and *Kat6a*$^{+/+}$ samples with respect to 87% of the genes differentially expressed at FDR < 0.05 between *Kat6a*$^{-/-}$ and *Kat6a*$^{+/+}$ E12.5 dorsal telencephalon (Fig 2C). The genes down-regulated in *Kat6a*$^{-/-}$ versus *Kat6a*$^{+/+}$ dorsal telencephalon were expressed in a diverse range of organs including the brain (Fig 2D), whereas those up-regulated were associated with expression in the brain only (Fig 2E). Of the genes differentially expressed in *Kat6a*$^{-/-}$ versus *Kat6a*$^{+/+}$ dorsal telencephalon (Table S5 (a); Fig 2F and G), down-regulated genes were predominantly annotated with gene ontology terms relating to regulation of transcription and translation with neurogenesis terms forming a smaller component (Fig S4E; Table S5 (b)). In contrast, genes up-regulated were more specifically associated with brain development, central nervous system development, synapse organisation, and neurogenesis (Fig S4F; Table S5 (c)). However, in both cases only very few genes formed the basis of the annotation (Table S5). Similarly, only very few genes supported the gene ontology term annotations of genes down-regulated in *Kat6a*$^{+/-}$ versus *Kat6a*$^{+/+}$ dorsal telencephalon (Fig S4G; Table S6 (b)). In contrast, genes

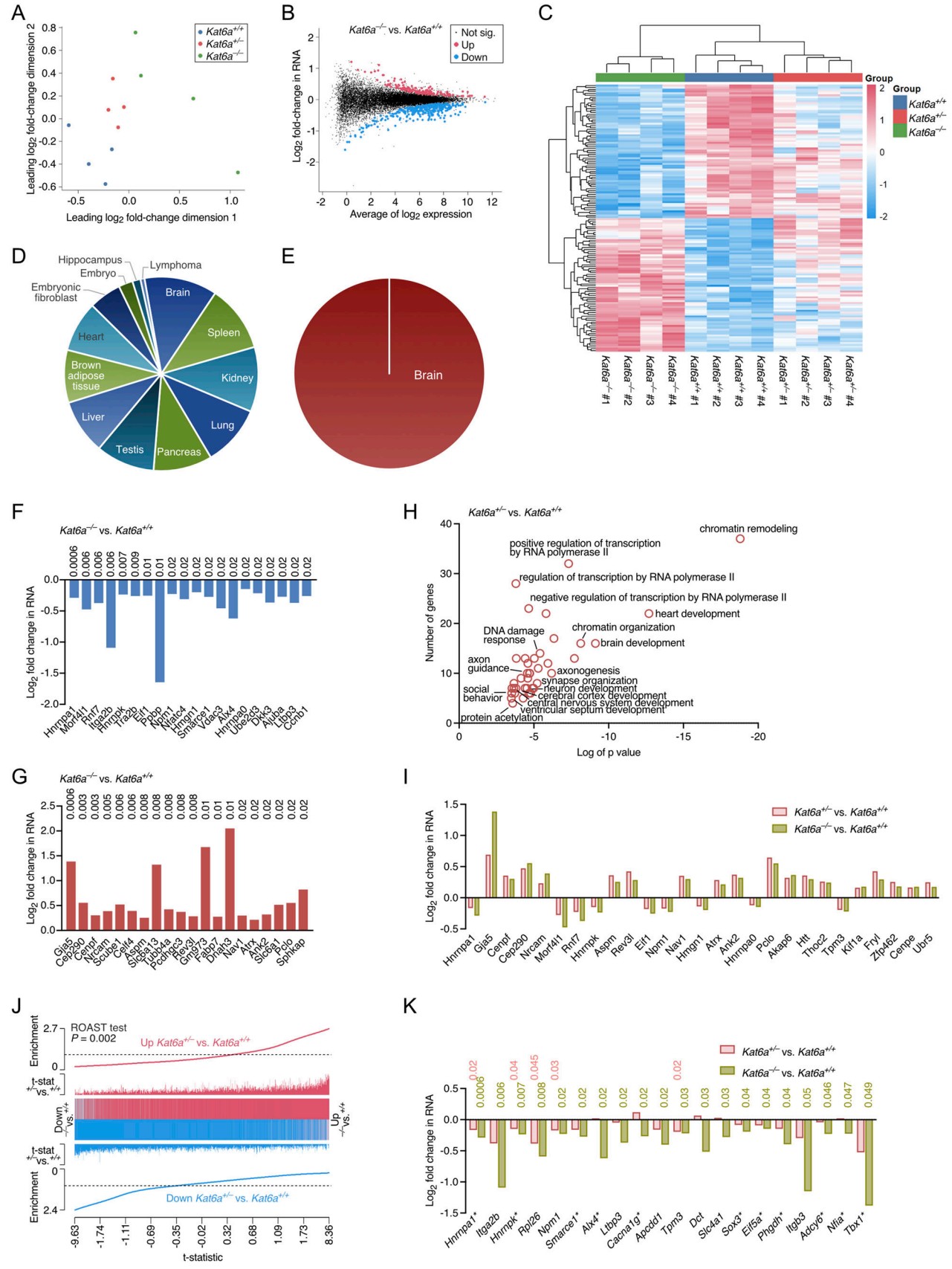

up-regulated in Kat6a$^{+/-}$ versus Kat6a$^{+/+}$ dorsal telencephalon were robustly associated with regulation of transcription, chromatin organisation, brain development, axonogenesis, synapse organisation, and neuron development (Fig 2H; Table S6 (c)), possibly indicating precocious neuronal differentiation. Genes that were differentially expressed in Kat6a$^{+/-}$ versus Kat6a$^{+/+}$ and in Kat6a$^{-/-}$ versus Kat6a$^{+/+}$ dorsal telencephalon were changed in the same direction (FDR < 0.05; Fig 2I). More generally, gene expression changes were highly correlated in the comparisons Kat6a$^{+/-}$ versus Kat6a$^{+/+}$ and Kat6a$^{-/-}$ versus Kat6a$^{+/+}$ telencephalon (P = 0.002; Fig 2J). Positive correlations were observed between gene expression changes in Kat6a$^{+/-}$ versus Kat6a$^{+/+}$ dorsal telencephalon and Kat6a$^{+/-}$ versus Kat6a$^{+/+}$ total adult hippocampus and CA3 pyramidal neurons (GSE261058; GSE261148 [Liu et al, 2024]) despite the diversity of tissue type and age (R$^2$ = 0.55 and 0.67, respectively; P = 0.0009 and 0.002, respectively; Fig S4H and I).

The genes down-regulated in Kat6a$^{-/-}$ versus Kat6a$^{+/+}$ dorsal telencephalon included genes that when mutated cause syndromes that include clinical features that also occur in ARTHS, including neurologic (11 genes), craniofacial (10 genes), musculoskeletal (11 genes), and cardiovascular defects (4 genes) (Table S7; Fig 2K). Five of the 20 human disorder–associated genes that were down-regulated in Kat6a$^{-/-}$ versus Kat6a$^{+/+}$ dorsal telencephalon were also significantly down-regulated in Kat6a$^{+/-}$ versus Kat6a$^{+/+}$ (FDR < 0.05; Fig 2K; Table S8).

These data suggested that loss of KAT6A reduced the expression of a range of genes required for normal development and maintenance of cognitive and other functions that are affected in individuals with ARTHS.

## Behavioural testing of Kat6a$^{+/-}$ and Kat6a$^{+/+}$ mice

To determine whether Kat6a$^{+/-}$ mice could be used as a model for the effects of loss of function of one allele of KAT6A on behaviour, learning and memory, and social interaction, we subjected Kat6a$^{+/-}$ and Kat6a$^{+/+}$ mice to a battery of behavioural tests. To establish their suitability for behavioural testing, we assessed visual acuity and physical abilities. Adult Kat6a$^{+/-}$ and Kat6a$^{+/+}$ mice displayed similar activity levels in their home cage (Fig S5A–D), similar visual function (Fig S5E), similar strength (Fig S5F), and similar fine motor coordination

(Fig S5G), suggesting that the mice would be capable of performing behavioural tests. Kat6a$^{+/-}$ mice spent more time in the centre of their home cage than Kat6a$^{+/+}$ mice (P = 0.003; Fig S5C), suggesting possibly a reduction in natural anxiety or a cognitive failure to recognise the risks associated with open spaces (further examined in additional tests below).

In addition, Kat6a$^{+/-}$ mice met six of eight physical and behavioural developmental milestones tested from birth to postnatal day 21 (P21) normally compared with Kat6a$^{+/+}$ littermate control mice (Fig S6A). Kat6a$^{+/-}$ mice displayed a 2.8-d delay in eye opening (P = 0.0003; Fig 3A) but were capable of negative geotaxis 3.7 d earlier than Kat6a$^{+/+}$ mice (P = 2 × 10$^{-5}$; Fig 3B).

## Kat6a$^{+/-}$ mice display reduced ultrasonic vocalisation

Individuals with ARTHS display mild to severe speech impairment (Kennedy et al, 2019; St John et al, 2022). We therefore assessed ultrasonic vocalisation (USV) in Kat6a$^{+/-}$ and Kat6a$^{+/+}$ mice in two settings: maternal separation–induced USV in postnatal day 4-, 8-, and 12-old mice and courtship USV in adult male mice. Kat6a$^{+/-}$ mice displayed reduced USV counts in both settings compared with littermate Kat6a$^{+/+}$ control mice (Figs 3C–E and S6B–F). The number of USV calls was reduced in both settings (P = 0.004–0.03; Fig 3C and D). In addition, adult Kat6a$^{+/-}$ males showed reduced vocalisation time (P = 0.04; Fig 3E). The percentage of calls in specific call categories (Yao et al, 2023) was largely unaffected, with only 1 or 2 of 10 call categories differentially represented at P4, P8, or P12 (p = 2 × 10$^{-5}$ to 0.02; Fig S6B–D and F).

## Kat6a$^{+/-}$ mice display reduced motor coordination

Kat6a$^{+/-}$ mice showed a reduction in motor coordination or strength in the rotarod test compared with Kat6a$^{+/+}$ mice (Fig 3F). Together, the normal grip strength (Fig S5B), the normal fine motor coordination (Fig S5C), and the lower performance on the rotarod (Fig 3F) suggest normal strength and a deficit in gross motor coordination in the rotarod test despite normal fine motor coordination. Alternatively, the lower performance on the rotarod may be due to a lack of anxiety associated with the prospect of falling from the rotarod.

---

**Figure 2. Loss of KAT6A affects expression of brain development genes.**
(A, B, C, D, E, F, G, H, I, J, K) RNA-sequencing data of dorsal telencephalon isolated from N = 4 Kat6a$^{+/+}$, 4 Kat6a$^{+/-}$, and 4 Kat6a$^{-/-}$ E12.5 embryos. Data were analysed as described in the Materials and Methods section under RNA-sequencing data analysis. Differences in gene expression with false discovery rate (FDR) < 0.05 were considered significant. **(A)** Multidimensional scaling plot of the leading gene expression differences between samples in pairwise comparisons of Kat6a$^{+/+}$, Kat6a$^{+/-}$, and Kat6a$^{-/-}$ dorsal telencephalon samples. **(B)** M (log ratio) and A (mean average) plot of genes differentially expressed in Kat6a$^{-/-}$ versus Kat6a$^{+/+}$ dorsal telencephalon. **(C)** Heatmap showing genes differentially expressed in the contrast of Kat6a$^{-/-}$ versus Kat6a$^{+/+}$ dorsal telencephalon but also displaying results for Kat6a$^{+/-}$ dorsal telencephalon. **(D)** Pie chart showing the tissue association of genes down-regulated in Kat6a$^{-/-}$ versus Kat6a$^{+/+}$ dorsal telencephalon. **(E)** Pie chart showing the tissue association of genes up-regulated in Kat6a$^{-/-}$ versus Kat6a$^{+/+}$ dorsal telencephalon. **(F)** Log$_2$ fold change in RNA levels of the top 20 genes down-regulated in Kat6a$^{-/-}$ versus Kat6a$^{+/+}$ dorsal telencephalon. FDRs are shown above the bars. **(G)** Log$_2$ fold change in RNA levels of the top 20 genes up-regulated in Kat6a$^{-/-}$ versus Kat6a$^{+/+}$ dorsal telencephalon. FDRs are shown above the bars. **(H)** Top 20 gene ontology terms (biological process) associated with genes up-regulated in Kat6a$^{+/-}$ versus Kat6a$^{+/+}$ dorsal telencephalon. **(I)** Log$_2$ fold change in RNA levels of the genes differentially expressed in Kat6a$^{-/-}$ versus Kat6a$^{+/+}$ and Kat6a$^{+/-}$ versus Kat6a$^{+/+}$ dorsal telencephalon, both FDR < 0.05. **(J)** Barcode plot showing a significant positive correlation (P = 0.002) between gene expression changes in Kat6a$^{-/-}$ versus Kat6a$^{+/+}$ and Kat6a$^{+/-}$ versus Kat6a$^{+/+}$ E12.5 dorsal telencephalon. The horizontal axis shows t-statistics for all genes in the Kat6a$^{-/-}$ versus Kat6a$^{+/+}$ dataset. The vertical lines represent genes in the Kat6a$^{+/-}$ versus Kat6a$^{+/+}$ dataset. Worms show the relative enrichment of the genes up-regulated (red) and down-regulated (blue) in the Kat6a$^{+/-}$ versus Kat6a$^{+/+}$ dataset. **(K)** Log$_2$ fold change in RNA levels of genes down-regulated in Kat6a$^{-/-}$ versus Kat6a$^{+/+}$ E12.5 dorsal telencephalon that are mutated in human disorders. The log$_2$ fold change in RNA levels is also shown for the comparison of Kat6a$^{+/-}$ versus Kat6a$^{+/+}$ E12.5 dorsal telencephalon. FDRs are shown above the bars.

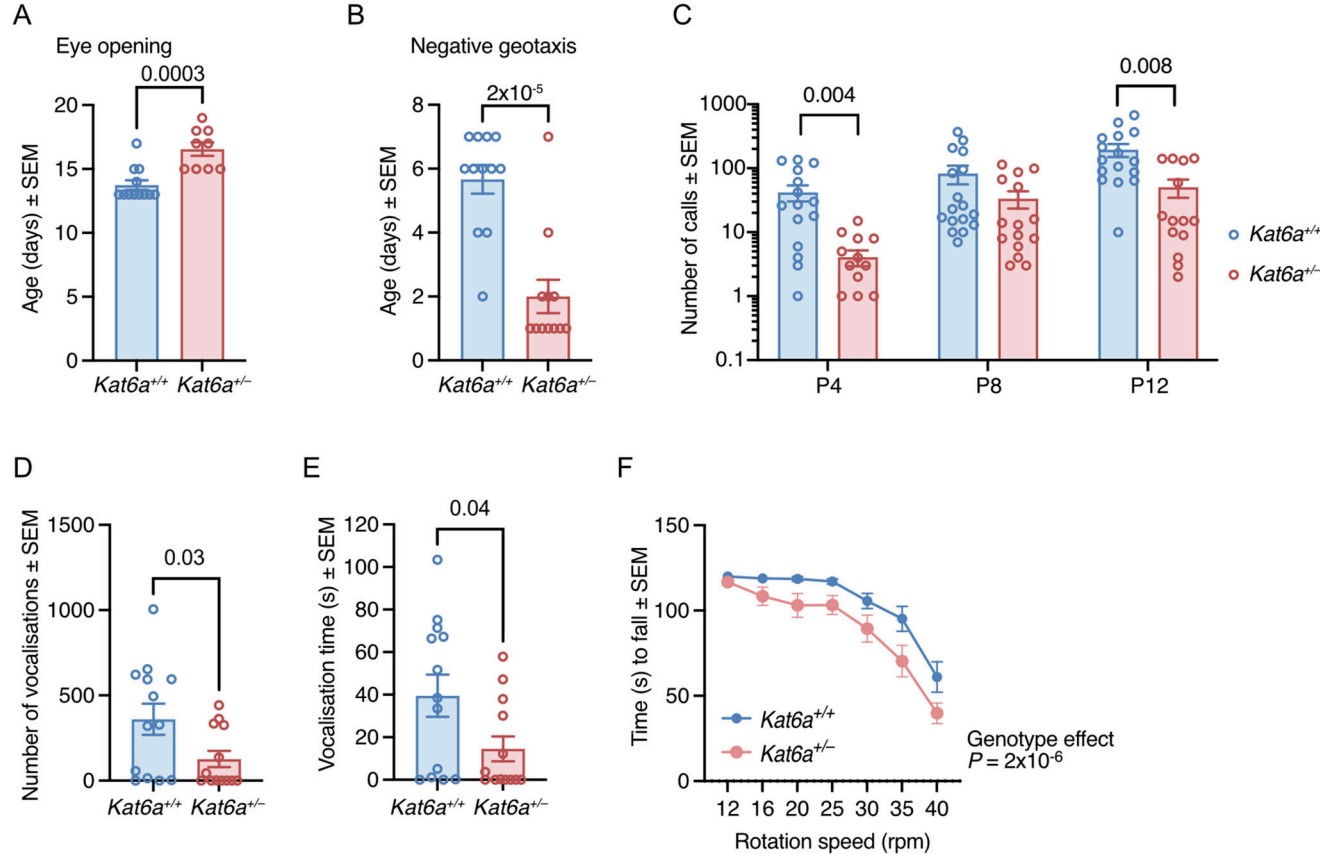

**Figure 3. Kat6a⁺/⁻ heterozygous mice display a reduction in ultrasonic vocalisation as pups and adults.**
**(A, B)** Developmental milestones eye opening (A) and negative geotaxis (B) were reached later (A) and earlier (B) by *Kat6a⁺/⁻* versus *Kat6a⁺/⁺* mice. **(C, D, E)** Assessment of ultrasonic vocalisation in *Kat6a⁺/⁻* versus *Kat6a⁺/⁺* mice induced by maternal separation at postnatal days 4, 8, and 12 (P4, P8, P12; (C)) and adult male vocalisation in response to the presence of a female mouse in oestrus (D, E). **(F)** Assessment of motor coordination and strength in *Kat6a⁺/⁻* versus *Kat6a⁺/⁺* mice using the rotarod. Effects of genotype, $P = 2 \times 10^{-6}$. N = 9–12 (A, B), 14–17 (C), 13 (D, E), and 17 (F) mice per genotype. Each circle in (A, B, C, D, E) represents an individual mouse. Data are displayed as the mean ± SEM and were analysed by an unpaired, two-tailed *t* test (A, B, D, E) and two-way ANOVA with Šídák's multiple comparison test (C, F).

### Kat6a⁺/⁻ mice display hyperactivity and a reduction in natural anxiety

To assess anxiety and activity levels, we assessed *Kat6a⁺/⁻* and *Kat6a⁺/⁺* mice in the large open field, the elevated zero maze, and the light/dark box. WT mice display a preference for the periphery near to the walls of the large open field, for the wall-enclosed sections of the elevated zero maze, and for the dark part of the light/dark box.

Kat6a⁺/⁻ mice displayed an increase in the distance travelled and speed in the large open field ($P$ = 0.02–0.03; Fig 4A–D). *Kat6a⁺/⁻* mice spent a similar percentage of time in the centre of the open field as *Kat6a⁺/⁺* mice (Fig 4E). *Kat6a⁺/⁻* mice similarly travelled a greater distance than *Kat6a⁺/⁺* mice in the elevated zero maze ($P$ = 0.001; Fig 4F and G). Interestingly, although *Kat6a⁺/⁻* mice spent a normal percentage of time in the centre of the open field (Fig 4E), they spent a larger percentage of time in the open sections of the elevated zero maze than *Kat6a⁺/⁺* mice ($P$ = 0.0001; Fig 4H) and also entered the open sections more frequently (p = 2 × 10⁻⁶; Fig 4I). Furthermore, *Kat6a⁺/⁻* mice spent

more time than *Kat6a⁺/⁺* mice in the light part of the light/dark box ($P$ = 0.03; Fig 4J and K). The behaviour of *Kat6a⁺/⁻* mice in the large open field and in the elevated zero maze suggests hyperactivity, and their behaviour in the elevated zero maze and the light/dark box suggests a reduction in natural anxiety or a cognitive failure to recognise the risks associated with open and light spaces.

### Kat6a⁺/⁻ mice show a deficit in learning and memory

Individuals with ARTHS display impaired cognition (Kennedy et al, 2019). To assess learning and memory, we tested *Kat6a⁺/⁻* and *Kat6a⁺/⁺* mice in the Y maze, in the novel object recognition test, and in the Barnes maze. The former two settings test if mice can distinguish between a familiar versus novel space or object, whereas the latter tests if they can learn to find a target location and develop an efficient strategy for the search.

The Y maze allows two tests assessing (a) working memory and (b) spatial memory. *Kat6a⁺/⁻* and *Kat6a⁺/⁺* mice performed equally well in the working memory test by visiting each arm of the three arms of

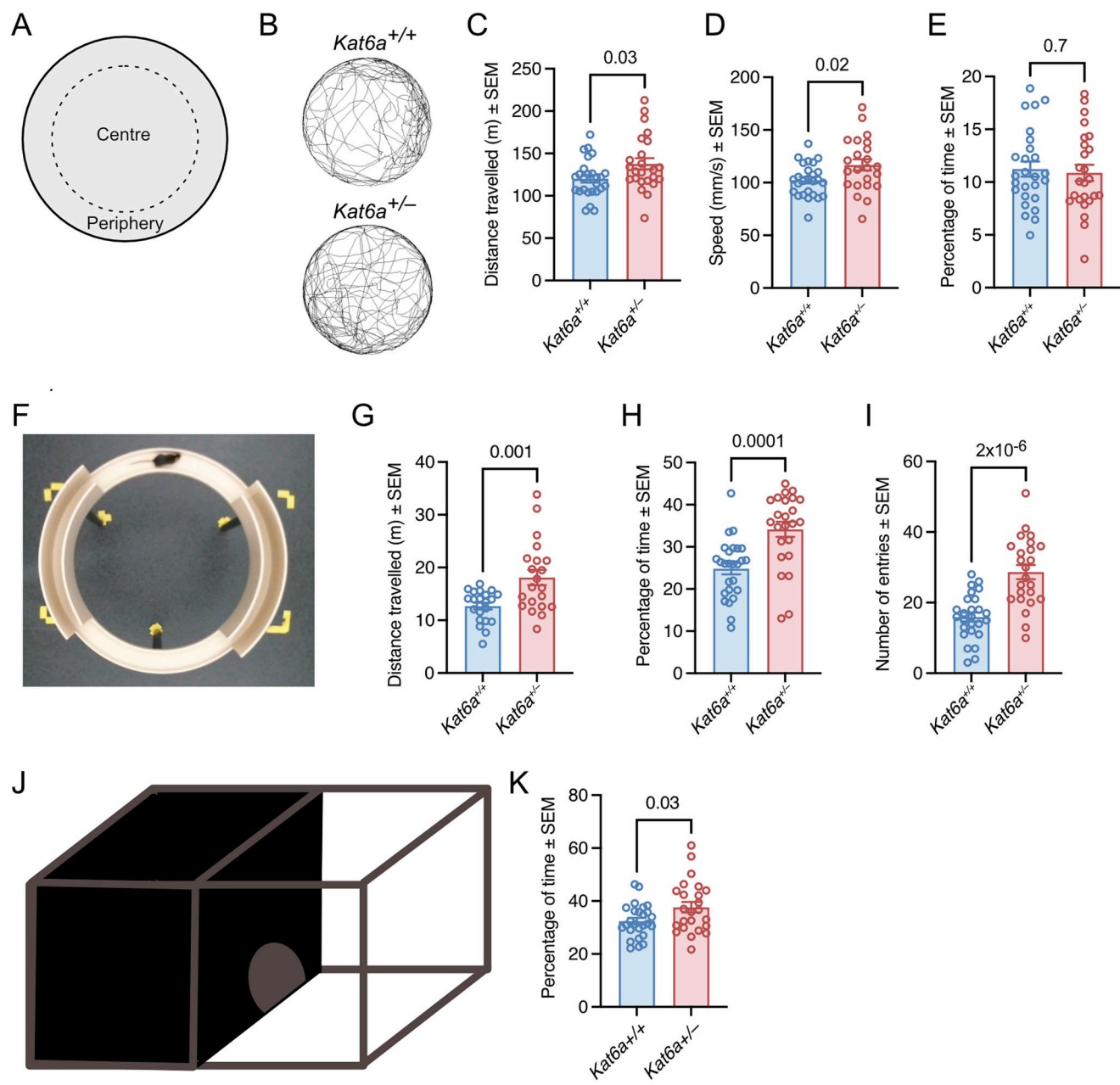

**Figure 4.** *Kat6a*$^{+/-}$ **heterozygous mice display hyperactivity and reduced natural anxiety.**
**(A)** Schematic drawing of the large open field indicating the periphery and the centre. **(B)** Representative traces of movements assessed by video camera and MouseMove software of *Kat6a*$^{+/-}$ and *Kat6a*$^{+/+}$ mice in the large open field. **(C, D, E)** Assessment of *Kat6a*$^{+/-}$ versus *Kat6a*$^{+/+}$ mice in a 20-min interval in the large open field determining total distance travelled (C), travel speed (D), and percentage of time spent in the centre (E). **(F)** View of an elevated zero maze from above. The open sections and the sections enclosed by walls can be discerned. **(G, H, I)** Assessment of *Kat6a*$^{+/-}$ versus *Kat6a*$^{+/+}$ mice in a 5-min interval in the elevated zero maze determining total distance travelled (G), percentage of time spent in the open sections (H), and number of entries into the open sections (I). **(J)** Schematic drawing of the light/dark box. **(K)** Percentage of time *Kat6a*$^{+/-}$ versus *Kat6a*$^{+/+}$ mice spent in the light section of the light/dark box. N = 21–26 mice per genotype. Each circle in (C, D, E, G, H, I, K) represents an individual mouse. Data are displayed as the mean ± SEM and were analysed by an unpaired, two-tailed *t* test.

the Y maze in sequence before revisiting one arm again in about 60% of the arm choices (Fig S7A). *Kat6a*$^{+/-}$ underwent more total number of entries into arms compared with the *Kat6a*$^{+/+}$ mice (*P* = 0.0006; Fig S7B), congruent with the hyperactivity shown in the large open field (Fig 4C and D) and the elevated zero maze (Fig 4G and I).

In the spatial memory test, *Kat6a*$^{+/+}$ mice showed the expected preference for the novel arm over the familiar arm with respect to time spent in the novel versus the familiar arm (*P* = 4 × 10$^{-6}$; Fig 5A and B) and when assessed by the discrimination index (*P* = 0.002; Fig 5C). In contrast, *Kat6a*$^{+/-}$ mice failed to display a preference for

the novel arm based on the time spent in the novel arm versus time spent in the familiar arm ($P$ = 0.2–0.3; Fig 5B) and this lack of preference was significantly different from the behaviour of the $Kat6a^{+/+}$ mice ($P$ = 0.0007; Fig 5C). However, when we examined the same data based on distance travelled (Fig 5D and E) and arm entries (Fig 5F and G), we found that $Kat6a^{+/-}$ mice did show a preference to travel a greater distance in the novel than the familiar arm ($P$ = $10^{-6}$; Fig 5D) and also entered the novel arm more frequently ($P$ = 0.0002; Fig 5F). Nevertheless, assessed by the discrimination index, the degree of preference displayed by $Kat6a^{+/-}$ mice was lower than in $Kat6a^{+/+}$ mice ($P$ = 0.006–0.05; Fig 5E and G).

In the novel object recognition test, $Kat6a^{+/+}$ mice showed the expected preference for the novel objects over the familiar object with respect to time spent with the objects ($P < 10^{-6}$; Fig 5H and I). $Kat6a^{+/-}$ mice also displayed a preference for the novel object (p = $4 \times 10^{-6}$; Fig 5I). However, assessed by the discrimination index, the degree of preference was lower in the $Kat6a^{+/-}$ compared with the $Kat6a^{+/+}$ mice ($P$ = 0.007; Fig 5J).

In the Barnes maze, both $Kat6a^{+/-}$ and $Kat6a^{+/+}$ mice found the target hole (1 of 20 holes available) equally fast (Fig S7C) and after equal numbers of non-target hole visits (Fig S7D) on testing days 1, 2, and 3. On testing day 4, $Kat6a^{+/-}$ mice arrived at the target hole later than $Kat6a^{+/+}$ mice ($P$ = 0.01; Fig S7C) and after significantly more prior visits of non-target holes ($P$ = 0.009; Fig S7D). $Kat6a^{+/-}$ mice displayed a greater latency to enter the target hole compared with $Kat6a^{+/+}$ mice ($P$ = 0.009; Fig S7E), consistent with a lower level of natural anxiety with respect to open spaces also observed in the elevated zero maze (Fig 4H) and the light/dark box (Fig 4K). To assess an elaborate learning process, we determined the search strategies used to arrive at the target hole. In the Barnes maze (Fig 5K), three strategies are distinguished: a random search, where holes are visited in a spatially random pattern (Fig 5L); a serial search, whereby a mouse moves to the periphery of the maze and then visits holes serially along the periphery (Fig 5M); a spatial search, where a mouse approaches the target hole directly with one visit of a directly adjacent hole allowed (Fig 5N). Over the 4 d of testing, $Kat6a^{+/+}$ mice increased the proportion of spatial searches ($P < 10^{-6}$) and decreased the proportion of random searches ($P < 10^{-6}$; Fig 5O). In contrast, $Kat6a^{+/-}$ mice failed to increase the use of the spatial search and continued the same search patterns for 4 d (Fig 5O).

Together, these data suggest that although $Kat6a^{+/-}$ mice can distinguish novel from familiar spaces and objects, they show a reduced preference for the novel situation. Importantly, $Kat6a^{+/-}$ mice seem incapable of learning a more complex spatial memory task. Accordingly, the results of the tests that suggested lack of natural anxiety (Fig 4I and K) may be due to cognitive deficits rather than a direct reduction in anxiety.

### $Kat6a^{+/-}$ mice show reduced interest in social interaction

Individuals with ARTHS display autism-like behaviours (Ng et al, 2024). To assess social interactions, we tested $Kat6a^{+/-}$ and $Kat6a^{+/+}$ mice in the three-chamber social interaction test. $Kat6a^{+/+}$ mice showed the expected preference for a cage containing a mouse over an empty cage ($P < 10^{-6}$; Fig 6A and B). $Kat6a^{+/-}$ mice

also displayed a preference for the mouse ($P$ = 0.004; Fig 6B). However, assessed by discrimination index, the degree of preference for the interaction with the mouse appeared to be lower in $Kat6a^{+/-}$ compared with the $Kat6a^{+/+}$ mice ($P$ = 0.02; Fig 6C), suggesting that $Kat6a^{+/-}$ mice mirror some of the sociability deficits of individuals with ARTHS.

### Treatment with acetyl-carnitine improves learning and memory in $Kat6a^{+/-}$ mice

Because histone acetylation is the catalytic function of KAT6A, it was possible that the reduction in histone acetylation levels observed in human cells with ARTHS KAT6A mutations (Fig 1C and D) and in the brain of mice lacking one allele of $Kat6a$ (Fig 1F and G) might contribute to the hyperactivity and cognitive and sociability deficits observed in individuals with ARTHS and in $Kat6a^{+/-}$ mice and that restoring histone acetylation levels might result in amelioration of the deficits. To explore this possibility, we treated $Kat6a^{+/-}$ and $Kat6a^{+/+}$ mice with the acetyl-donor, acetyl-L-carnitine (ALCAR). In mitochondria, acetyl-groups can be transferred from acetyl-CoA to carnitine to form acetyl-carnitine by carnitine-acyltransferase. In this form, mediated by carnitine, acetyl-groups can be exported from the mitochondria for use in other cell compartments, including the nucleus (Hesselink et al, 2016), where acetyl-groups are then available for acetylation processes, including for histone acetylation. Exogenous administration of acetyl-carnitine can similarly reach cell compartments and there serve as acetyl-donors. Because brain development continues after birth through adolescence (Stiles & Jernigan, 2010; Semple et al, 2013), we began treatment early, and because it was possible that there would be a continued requirement for histone acetylation throughout life, we continued treatment throughout the experiments. We treated litters of $Kat6a^{+/-}$ and $Kat6a^{+/+}$ mice with ALCAR or vehicle control by subcutaneous injection from postnatal days 14 to 28, and from then onwards via the feed (Fig 7A). H3K23ac levels were reduced in the brain and the spleen of vehicle-treated $Kat6a^{+/-}$ mice compared with $Kat6a^{+/+}$ mice ($P$ = 0.02–0.04; Fig 7B–E). After treatment with ALCAR, H3K23ac levels in the brain were no longer different between ALCAR-treated $Kat6a^{+/-}$ mice and vehicle- or ALCAR-treated $Kat6a^{+/+}$ mice (both $P$ = 0.3; Fig 7B and C) and were also not different in the spleen between ALCAR-treated $Kat6a^{+/-}$ mice and vehicle-treated $Kat6a^{+/+}$ mice ($P$ = 0.5; Fig 7D and E). H3K9ac levels in the brain or spleen were not affected by ALCAR treatment (Fig S8A–D). H3K14ac levels were increased by ALCAR treatment in the brain of $Kat6a^{+/-}$ mice ($P$ = 0.04; Fig S8E and F), but were unchanged in the spleen (Fig S8G and H).

ALCAR or vehicle treatment did not change the differences in body weight between $Kat6a^{+/-}$ and $Kat6a^{+/+}$ mice observed in the baseline assessment (Fig S8I cf. Fig 1I). ALCAR treatment did not improve the performance of the $Kat6a^{+/-}$ mice on the rotarod (Fig S8J cf. Fig 3F). Visual acuity (Fig S8K cf. Fig S5A) and grip strength (Fig S8L cf. Fig S5B) were similar to the baseline in $Kat6a^{+/-}$ and $Kat6a^{+/+}$ mice after ALCAR and vehicle treatment.

Vehicle-treated $Kat6a^{+/-}$ mice travelled a longer distance than $Kat6a^{+/+}$ mice in the large open field ($P$ = 0.02; Fig 7F), as seen in the baseline assessment (Fig 4C). In contrast, ALCAR-treated mice

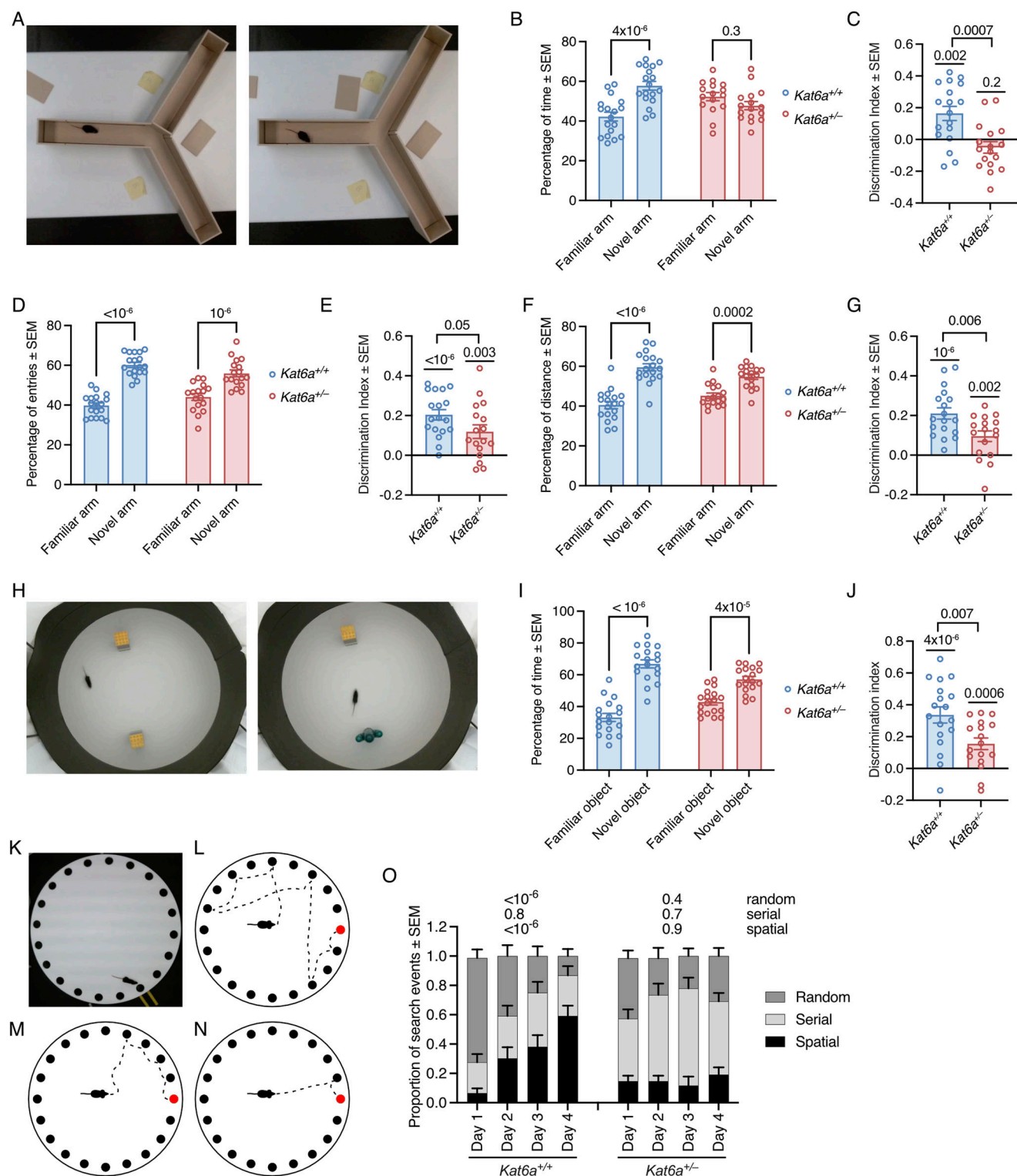

**Figure 5.** *Kat6a*[+/−] **heterozygous mice display reduced spatial strategy learning and memory.**
**(A)** View of a Y maze from above. One of the three arms is closed off in the first session (left image), which by opening it up becomes the "novel," unexplored arm in a second session (right image) of the spatial recognition memory test. **(B, C, D, E, F, G)** Assessment of *Kat6a*[+/−] versus *Kat6a*[+/+] mice in the Y maze spatial recognition memory test determining the percentage of time spent in the novel arm versus familiar arm of the Y maze (B) with the discrimination index for the novel arm (C), percentage of entries into the novel versus familiar arm (D) with the discrimination index (E), and the distance travelled in the novel versus familiar arm (F) with the discrimination index (G). **(H)** View of the open field set up for the novel object recognition test with two identical objects in the first session (left image) and one of the previous, now familiar objects and a "novel," unexplored object in the second session (right image). **(I, J)** Assessment of *Kat6a*[+/−] versus *Kat6a*[+/+] mice in the object recognition test determining the percentage of time spent examining the novel versus familiar object (I) with the discrimination index. (K) View of the Barnes maze of spatial learning and memory from above, which shows an elevated circular platform with 20 holes in the periphery, only one of which leads to an escape tunnel and

travelled a similar distance as vehicle- or ALCAR-treated $Kat6a^{+/+}$ mice (both $P$ = 1.0; Fig 7F). ALCAR or vehicle treatment had no effects on the proportion of time spent in the centre by mice of either genotype (Fig S8M).

Vehicle-treated $Kat6a^{+/-}$ and $Kat6a^{+/+}$ mice travelled a higher percentage of distance in the novel versus the familiar arm in the Y maze ($P$ = 3 × 10⁻⁶ to 9 × 10⁻⁵; Fig 7G), and unlike in the baseline assessment (Fig 5E), the discrimination index did not suggest that the preference for the novel arm was weaker in vehicle-treated $Kat6a^{+/-}$ than $Kat6a^{+/+}$ mice ($P$ = 1.0; Fig 7H). ALCAR treatment did not change the performance of the $Kat6a^{+/-}$ and $Kat6a^{+/+}$ mice in the spatial recognition test in the Y maze (Fig 7G and H).

Vehicle-treated $Kat6a^{+/-}$ mice travelled a longer total distance than $Kat6a^{+/+}$ mice in the novel object recognition test ($P$ = 0.006; Fig 7I), but after ALCAR treatment, the distance travelled by $Kat6a^{+/-}$ mice was not significantly different from vehicle- or ALCAR-treated $Kat6a^{+/+}$ mice ($P$ = 0.3–0.4; Fig 7G). Vehicle-treated $Kat6a^{+/-}$ and $Kat6a^{+/+}$ mice displayed a preference for the novel object (both $P$ < 10⁻⁶; Fig 7J), and unlike in the baseline assessment (Fig 5J), the discrimination index did not suggest that the preference for the novel object was weaker in the $Kat6a^{+/-}$ than the $Kat6a^{+/+}$ mice ($P$ = 0.4; Fig 7K). ALCAR treatment did not improve the performance of the $Kat6a^{+/-}$ mice in the novel object recognition test (Fig 7I–K).

The apparent improvement in performance of the vehicle-treated $Kat6a^{+/-}$ mice in the novel object recognition test over their performance in the baseline analysis, combined with the observation that $Kat6a^{+/-}$ mice do not thrive during the early postnatal period, induced us to hypothesise that vehicle treatment (saline i.p. for 2 wk followed by mash) may represent fluid or nutritional supportive treatments during postnatal development that might cause an improvement in recognition behaviour in $Kat6a^{+/-}$ mice. Therefore, we tested a range of potentially supportive treatments, specifically in the novel object recognition test, compared with the standard mouse pellets, which were used as control in the baseline analysis. We found that i. p. injection with saline or citrate solution improved novel object preference in $Kat6a^{+/-}$ mice when compared to standard pellets (Fig S8N and O). $Kat6a^{+/-}$ mice treated orally with a liquid veterinary product similar to Sustagen for humans (Di-Vetelact) displayed the tendency of an improvement, suggesting that fluid support, rather than nutritional support, might improve the novel object preference in $Kat6a^{+/-}$ mice.

The similarity in total distances travelled by ALCAR-treated $Kat6a^{+/-}$ and $Kat6a^{+/+}$ mice in the large open field and the novel object recognition test suggests that ALCAR treatment might ameliorate the hyperactivity observed in $Kat6a^{+/-}$ mice.

In the Barnes maze, vehicle- and ALCAR-treated $Kat6a^{+/-}$ and $Kat6a^{+/+}$ mice reached the target hole in a similar time and after a similar number of hole visits (Fig S8P and Q). As previously observed in the baseline assessment (Fig 5O), vehicle- and ALCAR-treated $Kat6a^{+/+}$ mice increased the proportion of searches with direct access to the target hole over the 4-d testing period ($P$ < 10⁻⁶–10⁻⁵; Fig 7L) and decreased the random searches ($P$ < 10⁻⁶–10⁻⁵; Fig 7L), while keeping the proportion of serial searches similar. As in the baseline assessment (Fig 5O), vehicle-treated $Kat6a^{+/-}$ mice failed to increase the proportion of the spatially direct approaches to the target hole over the 4-d testing period ($P$ = 0.4; Fig 7L). However, unlike in the baseline assessment (Fig 5O) vehicle-treated $Kat6a^{+/-}$ mice did show a decrease in the proportion of random searches ($P$ = 0.0003) and an increase in serial searches ($P$ = 0.005; Fig 7L). Remarkably, ALCAR-treated $Kat6a^{+/-}$ mice increased the proportion of the spatially direct approaches ($P$ = 0.0005), increased the proportion of serial searches ($P$ = 0.04), and decreased the proportion of random searches ($P$ < 10⁻⁶; Fig 7L).

Finally, we assessed the effects of treatment on sociability in the three-chamber social interaction test. Vehicle- and ALCAR-treated $Kat6a^{+/+}$ mice displayed the expected preference for interaction with a mouse as compared to an empty cage ($P$ = 4 × 10⁻⁵ to 0.0002; Fig 7M). Unlike in the baseline assessment, where $Kat6a^{+/-}$ mice displayed a preference for interacting with a mouse over an empty cage (Fig 6B), albeit weaker than the preference of the $Kat6a^{+/+}$ mice (Fig 6C), vehicle-treated $Kat6a^{+/-}$ mice showed no significant preference for the mouse over an empty cage ($P$ = 0.2; Fig 7M) and ALCAR treatment had no effects (Figs 7M and S8R).

Taken together, our data suggest that treatment with ALCAR resulted in a reduction of the hyperactivity and an improvement of spatial strategy learning in $Kat6a^{+/-}$ mice, but did not improve sociability.

# Discussion

In this study, we have shown that four of six ARTHS mutations tested caused a reduction in H3K23ac in human cells. Similarly, heterozygous loss of $Kat6a$ in mice caused a reduction in H3K23ac in mouse brain and peripheral white blood cells. We showed that treatment of $Kat6a^{+/-}$ mice with ALCAR resulted in H3K23ac levels that were not significantly different from WT levels in the brain and in the spleen. In a battery of behavioural tests, $Kat6a^{+/-}$ mice displayed hyperactivity, learning and memory, and sociability defects, without having obvious anatomical or histological anomalies of the brain or cultured cortical neurons. In contrast, we did observe changes in gene expression in the developing $Kat6a^{+/-}$ and $Kat6a^{-/-}$ forebrain where genes involved in the regulation of transcription in the brain were affected, and genes that are mutated in a range of neurological disorders. Remarkably, the hyperactivity and learning defects of $Kat6a^{+/-}$ mice were ameliorated by treatment with ALCAR. The sociability defects were not improved by treatment. Our data suggest that a subset of individuals with ARTHS may benefit from treatment elevating histone acetylation levels.

chamber. **(L, M, N)** Schematic drawings of the search strategies mice display in the Barnes maze to find the target hole (target hole indicated in red), namely, random search (L), serial search (M), and direct approach of the target hole termed "spatial search" (N). **(O)** Assessment of the combination of search strategies used by $Kat6a^{+/-}$ versus $Kat6a^{+/+}$ mice to find the target hole in the Barnes maze over the 4-d training and test period. N = 16–18 (B, C, D, E, F, G), 18 (I, J), and 17–19 (O) mice per genotype. Each circle in (B, C, D, E, F, G, I, J) represents an individual mouse. Data are displayed as the mean ± SEM and were analysed by two-way ANOVA with Šídák's multiple comparison test (B, D, F, I, O), one-sample $t$ test compared with 0 as the theoretical mean, and unpaired, two-sided $t$ test (C, E, G, J).

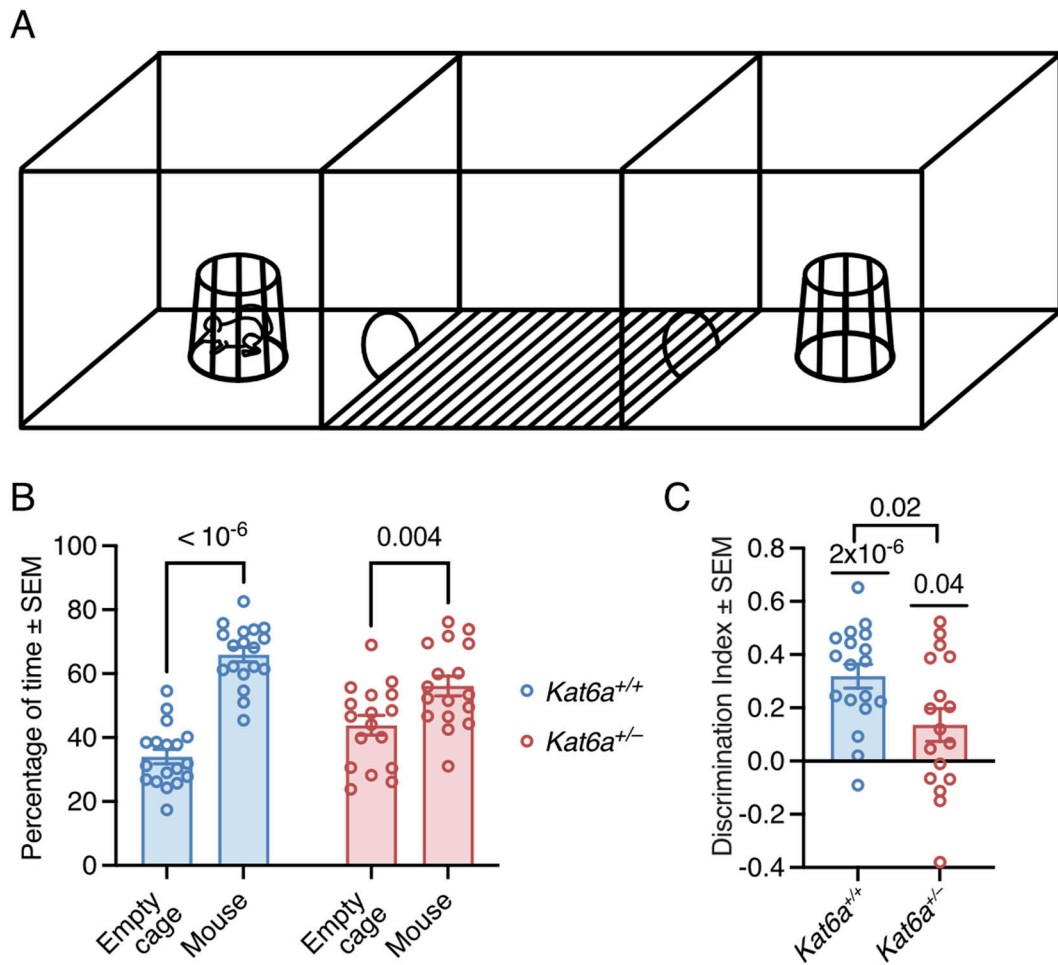

**Figure 6. _Kat6a_$^{+/-}$ heterozygous mice display reduced sociability.**
**(A)** Schematic drawing of the three-chamber social interaction test. **(B, C)** Assessment of _Kat6a_$^{+/-}$ versus _Kat6a_$^{+/+}$ mice in the three-chamber social interaction test determining the percentage of time spent with the mouse versus the empty cage (B) with the discrimination index (C). N = 17–18 mice per genotype. Each circle in (B, C) represents an individual mouse. Data are displayed as the mean ± SEM and were analysed by two-way ANOVA with Šídák's multiple comparison test (B) and one-sample _t_ test compared with 0 as the theoretical mean, as well as an unpaired, two-sided _t_ test (C).

Similar to our findings, Liu et al (2024) observed that mice lacking exon 3 of the _Kat6a_ gene displayed a deficit in learning and memory manifested as lack of preference for a novel object in the novel object recognition test and an inferior performance in the Morris water maze, which tests similar abilities as the Barnes maze used in our study. Likewise, they found a reduction in natural anxiety as observed in the elevated plus maze, which is similar to our results in the elevated zero maze. A notable difference between the two studies is that we observed hyperactivity and a sociability deficit in _Kat6a_$^{+/-}$ mice, whereas Liu and co-workers did not. Liu and colleagues did not test treatment options.

Our E12.5 RNA-sequencing data revealed a large number of genes up-regulated (282) in the absence of one allele of _Kat6a_ in the dorsal telencephalon, the cerebral cortex precursor, whereas more subtle effects in the foetal (our E16.5 data) and adult stage tissue (Liu et al, 2024) were seen. Interestingly, differentially expressed genes in _Kat6a_$^{+/-}$ versus _Kat6a_$^{+/+}$ adult hippocampus (Liu et al, 2024) correlated positively with the top changes in our

_Kat6a_$^{+/-}$ versus _Kat6a_$^{+/+}$ E12.5 dorsal telencephalon. The up-regulated genes in the _Kat6a_$^{+/-}$ E12.5 telencephalon were robustly associated with neuron development and maturation, such as axon guidance and synapse organisation. We speculate that this finding may reflect premature neuronal maturation perhaps at the expense of neuronal precursor population expansion, which might relate to the cognitive and social deficits.

Liu and co-workers observed impaired synaptic structure and plasticity in the hippocampal CA3 region (Liu et al, 2024). Of 90 genes differentially expressed in the CA3 region, they focussed on _Rspo2_ (_r-spondin 2_) and show that restoring _Rspo2_ gene expression via adenovirus injection rescues learning-associated deficits in _Kat6a_ mutant mice. It is worth noting that KAT6A is one of only nine nuclear lysine acetyltransferases with a defined acetyltransferase domain, nine proteins that are collectively responsible for all histone acetylation and other protein acetylation in the nucleus. Accordingly, our data suggest that hundreds of

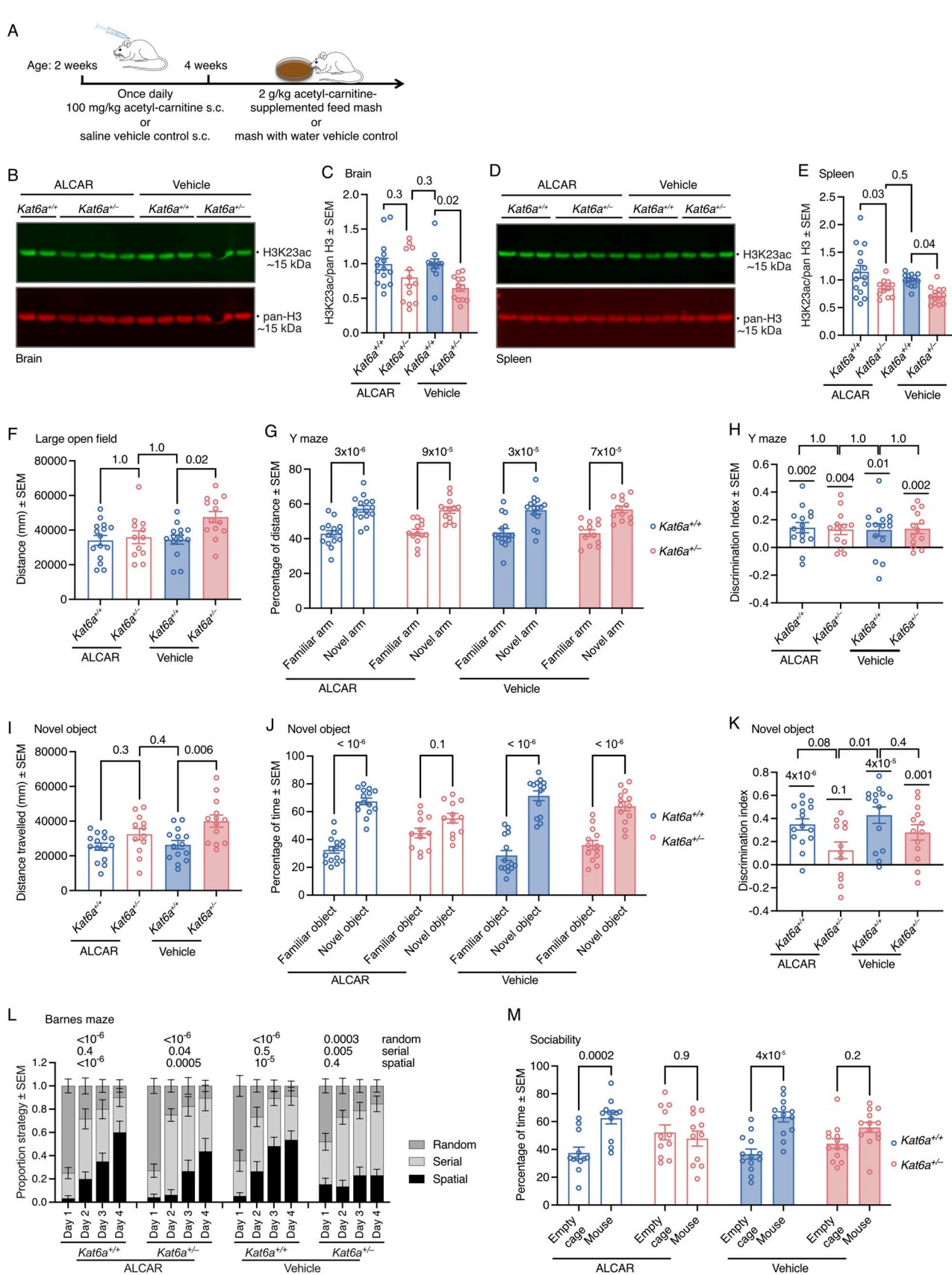

genes depend on KAT6A for their normal expression levels (and histone acetylation). It is therefore likely that *Rspo2* is only one of many KAT6A target genes.

KAT6A and KAT6B are two closely related paralogous histone acetyltransferases; indeed, the overexpression of KAT6B can rescue all lethal developmental defects in *Kat6a*$^{-/-}$ embryos (Bergamasco et al, 2025). Like mutations in *KAT6A*, mutation of one allele of the *KAT6B* gene also causes cognitive impairment syndromes (Clayton-Smith et al, 2011; Kraft et al, 2011; Campeau et al, 2012). Similar to the improvement observed in the learning and memory in *Kat6a*$^{+/-}$ mice after treatment with ALCAR observed here, a restoration of learning and memory in *Kat6b*$^{+/-}$ mice in response to ALCAR treatment was reported previously (Bergamasco et al, 2024). However, unlike the *Kat6a*$^{+/-}$ mice, the *Kat6b*$^{+/-}$ mice also responded with an improvement in sociability, preferring the company of a mouse over an empty cage after ALCAR treatment.

Four factors are important for the consideration of whether treatment increasing histone acetylation levels after a genetic diagnosis after birth might be of benefit. First, it would be important to determine for each individual with ARTHS if their personal *KAT6A* mutation caused a reduction in histone acetylation. Our data in mice suggest that peripheral white blood cells could be used to assess histone acetylation levels. Second is the observation that although brain development continues after birth (Stiles & Jernigan, 2010; Semple et al, 2013), it is most dynamic during the first few years of life. It is therefore likely that it would be beneficial to commence treatment early. Third, the *KAT6A* gene is expressed in the adult brain (Kraft et al, 2011; Dillman et al, 2013). It is therefore possible that KAT6A continues to be required for normal brain function throughout life, such that continued treatment may be beneficial. Fourth, ALCAR is used widely orally as a nutritional supplement, and therefore, information on its tolerability is available reducing the risks one needs to consider in association with any treatment.

Brain development continues after birth into early adulthood (Stiles & Jernigan, 2010; Semple et al, 2013). The brain of a new-born infant is approximately one third of the size of an adult brain (Holland et al, 2014), and a child reaches 95% of the adult brain size at 7–11 yr of age (Caviness et al, 1996), reflective of substantial growth in the years after birth. Although neurons are formed predominantly before birth, glial cells and myelination continue after birth. Pre- and postnatal brain development results in an overabundance ("developmental exuberance") of neurons, glia, neurites, and synapse connectivity (Innocenti & Price, 2005), which in pre- and postnatal life are consolidated by an activity-dependent mechanism or pruned. The cerebral grey matter reaches its peak at 10–12 yr of age in the frontal and parietal lobe and at 17 yr of age in the temporal lobe and thereafter declines in size, whereas the white matter increases at least until 22 yr of age (Giedd et al, 1999). Post-adolescence changes in the frontal cortex have also been observed (Sowell et al, 1999). The volume of brain regions declines throughout adult life with the exception of the cerebral white matter, which is at its highest levels between 30 and 50 yr of age before also declining (Jernigan & Gamst, 2005; Lebel & Beaulieu, 2011). Synapse density peaks at approximately twice the adult levels (e.g., between 3.5 and 7 yr of age in the prefrontal cortex), before it is reduced by pruning (Huttenlocher & Dabholkar, 1997). Based on this extended timeline of brain development and the continued expression of the *Kat6a* gene in the prenatal to adult brain, it is possible that an effective therapeutic intervention may be beneficial throughout life.

The mechanism by which ALCAR might improve hyperactivity and learning deficits might lie with its ability to act as an acetyl-donor for histone acetylation. However, in addition, ALCAR can contribute an acetyl-group to the formation of the neurotransmitter acetylcholine (Dolezal & Tucek, 1981). Acetylcholine is implicated in learning and memory, with increased acetylcholine levels associated with hippocampal-dependent learning (Chang & Gold, 2003). Furthermore, ALCAR has an antidepressant effect through epigenetic induction of a glutamatergic receptor (Nasca et al, 2013), an anxiolytic effect in rodents (Wang et al, 2015), leads to an increase in the neurotransmitters noradrenaline and serotonin in the cortex (Smeland et al, 2012), and can contribute to GABA synthesis (Scafidi et al, 2010). Given the many roles that ALCAR has in brain function, it is not possible to unambiguously attribute the improvement in hyperactivity and learning performance in *Kat6a*$^{+/-}$ mice to any one function of ALCAR. It is also unclear why ALCAR improved hyperactivity and learning, but not sociability deficits. This should not distract from the improvement in hyperactivity and learning observed after ALCAR treatment.

ARTHS presents with characteristic and variably occurring anomalies. The variable anomalies tended to be more common in individuals with truncating mutations in the last two *KAT6A* exons (exons 16–17) compared with earlier exons (1–15) (Kennedy et al, 2019). This divergence was particularly pronounced with respect to microcephaly

**Figure 7. Treatment with acetyl-carnitine increases H3K23 acetylation, reduces hyperactivity, and improves spatial strategy learning and memory in *Kat6a*$^{+/-}$ heterozygous mice.**

**(A, B, C, D, E, F, G, H, I, J, K, L, M)** *Kat6a*$^{+/-}$ and *Kat6a*$^{+/+}$ mice were treated with acetyl-carnitine versus vehicle control from 2 wk of age and subjected to behavioural testing in adulthood. **(A)** Schematic representation of the treatment regime. **(B, C, D, E)** H3K23 acetylation levels in *Kat6a*$^{+/-}$ versus *Kat6a*$^{+/+}$ mice after treatment with acetyl-carnitine versus vehicle control in the brain (B, C) and spleen (D, E) assessed by Western blot (representative blots shown, (B, D)) and densitometry (C, E). 1 µg (B) or 0.5 µg (D) of acid-extracted protein was loaded per lane. Each lane contains protein from one individual mouse. **(F)** Distance travelled by *Kat6a*$^{+/-}$ versus *Kat6a*$^{+/+}$ mice in a 5-min interval in the large open field after treatment with acetyl-carnitine versus vehicle control. **(G, H)** Distance travelled (G) with the discrimination index (H) by *Kat6a*$^{+/-}$ versus *Kat6a*$^{+/+}$ mice in the arms of the Y maze for spatial recognition memory after treatment with acetyl-carnitine versus vehicle control. **(I, J, K)** Distance travelled (I) and time spent examining objects (J) with the discrimination index (K) by *Kat6a*$^{+/-}$ versus *Kat6a*$^{+/+}$ mice in the novel object recognition test after treatment with acetyl-carnitine versus vehicle control. **(L)** Spatial strategy learning and memory in *Kat6a*$^{+/-}$ versus *Kat6a*$^{+/+}$ mice in the Barnes maze after treatment with acetyl-carnitine versus vehicle control. **(M)** Social preference of mouse versus empty cage of *Kat6a*$^{+/-}$ versus *Kat6a*$^{+/+}$ mice in the three-chamber social interaction test after treatment with acetyl-carnitine versus vehicle control. N = 12–14 (C, E), 12–15 (F, G, H, I, J, K, L), and 11–13 (M) mice per genotype. Each circle in (C, E, F, G, H, I, J, K, M) represents an individual mouse. Data are displayed as the mean ± SEM and were analysed by one-way ANOVA with Tukey's multiple comparison test (C, E, F, I), two-way ANOVA with Šidák's multiple comparison test (G, J, M), one-sample *t* test compared with 0 as the theoretical mean, as well as unpaired, two-sided *t* test (H, K), and two-way ANOVA with Benjamini and Hochberg correction for multiple testing (L).

(7.3-fold more frequent in individuals with late truncations), sleep disturbance (2.8-fold), constipation (2.6-fold), cardiac defects (2.5-fold), and frequent infections (2.3-fold) (Kennedy et al, 2019), suggesting a possible dominant negative effect of a truncated protein on the normal KAT6A protein produced by the remaining healthy allele (and possibly on the related KAT6B protein). In addition, late truncations are associated with greater cognitive and speech impairment (Kennedy et al, 2019; St John et al, 2022). It is possible that mRNA produced by alleles with truncating mutations in KAT6A exons 1 to 15 is subject to nonsense-mediated mRNA decay, which would render these mutations similar to a null allele, as we generated in our Kat6a mice. Indeed, our data in HEK293T cells support this hypothesis. Based on our HEK293T cell data, it is possible that individuals with truncating mutations in KAT6A exons 1–15 are more likely to have a reduction in H3K23ac and would be more likely to benefit from treatment that increases histone acetylation levels.

The hyperactivity of Kat6a⁺/⁻ mice was pronounced and was observed in several behavioural tests, and the spatial strategy learning and memory defect observed in Kat6a⁺/⁻ mice in the Barnes maze was obvious. Both were also clearly and significantly improved after ALCAR treatment. In contrast, the spatial novelty and object novelty recognition memory tests only returned subtle differences between Kat6a⁺/⁻ and Kat6a⁺/⁺ mice, which complicated the interpretation of the ALCAR treatment experiments. Indeed, in some of these tests the reduced preference for novelty observed in Kat6a⁺/⁻ versus Kat6a⁺/⁺ mice in the baseline analysis was not present in the vehicle-treated control group during ALCAR testing. The vehicle-treated group received daily saline injections from 14 to 28 d of age and were then fed mash without ALCAR. We hypothesised that a reason for a diminished difference in preference for novelty between Kat6a⁺/⁻ versus Kat6a⁺/⁺ mice might be that the saline injections and mash provide a supportive treatment that by itself improved the previously observed failure to thrive in Kat6a⁺/⁻ mice, thus also improving their performance in these simpler learning and memory tests. Indeed, we found that several liquid supportive treatments provided a benefit to Kat6a⁺/⁻ mice in the object recognition memory test, suggesting that liquid supportive treatment in the early postnatal phase, when Kat6a⁺/⁻ mice fail to thrive, might improve their development.

Overall, we have shown that Kat6a⁺/⁻ mouse model mimics the cognitive impairment, hyperactivity, and sociability deficits of individuals with ARTHS and that the cognitive impairment and hyperactivity can be improved by treatment with an acetyl-donor. Our findings suggest that treatment with ALCAR might be beneficial for a subset of individuals with ARTHS, once individually a deficiency in histone acetylation levels has been established.

# Materials and Methods

### Ethics approval

Experiments were approved by the Walter and Eliza Hall Animal Ethics Committee and were conducted in accordance with the NHMRC Australian code for the care and use of animals for scientific purposes.

### Human cells

HEK293T cells were supplied by M. Herold (Walter and Eliza Hall Institute of Medical Research).

### Homologous recombination of the human KAT6A locus

ARTHS-causing mutations were generated in HEK293T cells using CRISPR/Cas9 and homology-directed repair (HDR), as described previously (Jacobi et al, 2017). The Alt-R CRISPR HDR Design Tool (IDT; https://sg.idtdna.com/pages/tools/alt-r-crispr-hdr-design-tool) was used to design custom guide RNAs and single-stranded HDR donor DNA oligonucleotides to generate 6 KAT6A ARTHS mutations, as well as control cells that retained the WT KAT6A sequence but carried a novel EcoR1 restriction enzyme cut site in the AAVS1 locus (Table S1). A ribonucleoprotein (RNP) approach with Lipofectamine CRISPRMAX Cas9 Transfection Reagent (CMAX00015; Thermo Fisher Scientific) transfection was used for the c.235C>T (p.R79*) and c.3055C>T (p.R1019*) KAT6A mutations, following a modified Alt-R CRISPR-Cas9 System protocol (IDT). Briefly, custom crRNAs were complexed with trRNAs to form 1 μM gRNAs. Per HDR reaction, 3.75 μl of 1 μM gRNA, 0.125 μl of 5 μg/μl TrueCut Cas9 v2 (A36498; Thermo Fisher Scientific), and CRISPRMAX Cas9 Plus Reagent (CMAX00015; Thermo Fisher Scientific) were combined with 57.125 μl Gibco Opti-MEM (31985088; Thermo Fisher Scientific). After a 5-min incubation at RT, 3.75 μl of the appropriate 0.3 μM ssDNA repair template was added and incubated for another 5 min. Next, 59.5 μl Opti-MEM, 3 μl CRISPRMAX transfection reagent (CMAX00015; Thermo Fisher Scientific), and 0.55 μl Alt-R HDR Enhancer v2 (10007910; IDT) were added to the RNP complexes and incubated for 20 min at RT. HEK293T cells were diluted to 200,000 cells/ml in Gibco DMEM (11885084; Thermo Fisher Scientific) with 10% FBS and without antibiotics. In a 48-well plate, 250 μl diluted cells (50,000 cells/well) were combined with 125 μl of the transfection complex mixture and incubated for 24 h before removing RNP medium and replacing with DMEM with 10% FBS and 100 units/ml penicillin/streptomycin (15140122; Thermo Fisher Scientific). A ribonucleoprotein (RNP) approach delivered by Nucleofection (V4XC-2032; Lonza) was used for all other KAT6A mutations to increase the rate of HDR, following a modified Alt-R CRISPR-Cas9 System protocol (IDT). Briefly, custom crRNAs were complexed with trRNAs to form 50 μM gRNAs. Per HDR reaction, 3 μl of 50 μM gRNA and 2 μl of 5 μg/μl TrueCut Cas9 v2 (A36498; Thermo Fisher Scientific) were incubated together for 20 min. The resulting 5 μl of RNP complex was combined with 1.2 μl of 100 μM ssDNA repair template, 1.2 μl IDT electroporation enhancer (1075916; IDT), and 1 million HEK293T cells in 20 μl SF solution and supplement (V4XC-2032; Lonza). 25 μl was transferred into the Nucleofector cassette and electroporated using the CA-189 protocol (Zou et al, 2021). Cells were removed into prewarmed Gibco DMEM (11885084; Thermo Fisher Scientific) with 10% FBS containing 1.7 μl HDR Enhancer (10007921; IDT) per 1 ml media and incubated for 24 h before replacing media with DMEM with 10% FBS and 100 units/ml penicillin/ streptomycin (15140122; Thermo Fisher Scientific). Editing efficiency was determined on bulk-edited cell populations by targeted PCR of the Cas9-targeted region and secondary PCR using overhang sequences (Table S1), followed by Illumina MiSeq, as previously described (Aubrey et al, 2015). MiSeq results for the ARTHS mutations

generated are displayed in Table S2. The intended ARTHS mutation was dominantly represented in MiSeq reads (Table S3).

### RT–qPCR

RNA was isolated from HEK293T cells or mouse adult cortex using an RNA extraction kit (RNeasy Mini, 74106; QIAGEN). RNA was quantitated using a high-sensitivity RNA tape (5067-5579; Agilent, and 4200 TapeStation; Agilent), and 150 ng RNA was used for cDNA synthesis using a cDNA synthesis kit (SuperScript IV cDNA Synthesis Kit, 18091050; Thermo Fisher Scientific) according to the manufacturer's instructions. cDNA was amplified using a real-time PCR machine (QuantStudio, Thermo Fisher Scientific) with primers shown in Table S9.

### Mice

*Kat6a* heterozygous mice (*Kat6a*$^{+/-}$ [Voss et al, 2009]) were maintained on a *C57BL/6* background. The *Kat6a* null allele was generated as described previously, with the deletion of 5 exons (exons 4–8 of 17 exons in NM_001364449.1 and exons 5–9 of 18 exons in NM_001081149.2) resulting in a premature stop codon in the catalytic domain (Voss et al, 2009). Mice were housed in groups of 2–6 in cages with HEPA filtering with food (γ-irradiated feed, Barastoc; Ridley AgriProducts) and sterilised water provided ad libitum. Mice were held in a 14-h light and 10-h dark cycle. Embryos and foetuses were recovered after timed matings with noon of the day on which the vaginal plug was observed designated embryonic day 0.5 (E0.5). *Kat6a*$^{+/-}$ mice were intercrossed to obtain *Kat6a*$^{+/+}$, *Kat6a*$^{+/-}$, and *Kat6a*$^{-/-}$ E12.5 embryos. *Kat6a*$^{+/-}$ mice were crossed to WT *C57BL/6* mice to obtain E16.5 *Kat6a*$^{+/+}$ and *Kat6a*$^{+/-}$ foetuses and adult mice. A three-way PCR was used for genotyping (Table S10).

Mice were used for experiments in order of birth or embryo or foetal recovery. Male and female mice were used in order of birth. For the RNA-sequencing experiments, two males and two females per genotype were used.

#### *Acetyl-carnitine treatment of mice*
Cohorts of 12 mice born within 48 h of each other were divided into two groups undergoing a long-term treatment schedule. From postnatal days 14 to 28, mice were injected subcutaneously once per day with 100 mg/kg acetyl-L-carnitine (ALCAR; O-acetyl-L-carnitine, #A6706; Sigma-Aldrich) in saline (pH adjusted to 6.0 with sodium hydroxide) or saline vehicle control. This ALCAR regime was based on previous work showing behavioural improvements after treatment with ALCAR (Schaevitz et al, 2012; Bergamasco et al, 2024). From postnatal day 29 onwards, mice were administered a mash made of feed powder and water with or without 2 g ALCAR per kg feed. At 6 wk of age, mice underwent a battery of behavioural tests (described in detail below) while treatment continued. After all tests were completed, spleen and brain were collected for further analysis.

#### *Treatment of mice with citrate, saline, Di-Vetelact, or standard mouse feed pellets*
For citrate treatment, mice were injected intraperitoneally once per day with 77 mg/kg citrate solution (made up of 693 mg trisodium citrate, 77 mg citric acid, and 600 mg sodium in 100 ml H$_2$O) from postnatal days 14 to 28, along with standard mouse feed pellets and

drinking water. From 4 to 16 wk of age, mice were treated with the addition of 14.193 g sodium citrate and 12.872 g citric acid per litre to the drinking water, along with standard mouse feed pellets.

For saline treatment, mice were injected intraperitoneally once per day on postnatal days 14–28, along with standard mouse feed pellets and drinking water, followed by normal pellets and drinking water.

For Di-Vetelact treatment, mice were given Di-Vetelact ad libitum (Australian Business Number 41 164 057 792; Lillelund) according to the manufacturer's instructions, along with standard mouse feed pellets and drinking water.

Standard mouse feed pellets and drinking water were supplied ad libitum as a control treatment.

### Foetal cortical neuron isolation and culture

The dorsal forebrain was dissected from E16.5 foetuses, incubated in trypsin/EDTA (1006132; Sigma-Aldrich) on ice for 30 min, then at 37°C for 5 min, rinsed in 2% BSA in PBS, and dissociated with gentle pipetting. Samples were centrifuged for 10 min at 200*g* at 4°C, the supernatant was removed, and the cell pellet was resuspended in 2% BSA in PBS and the suspension was centrifuged for 10 min at 200*g* at 4°C. Cells were resuspended in Neurobasal medium (21103049; Life Technologies) supplemented with 2% B27 supplement (17504044; Life Technologies) and 500 μM Gluta-MAX supplement (35050061; Life Technologies), then passed through a 40-μM sieve. Cell preparations from each foetus were plated on one 6-cm dish previously coated with poly-D-lysine (P6407-5MG; Sigma-Aldrich) and grown at 37°C in 5% CO$_2$.

### Postnatal cortical neuron isolation and culture

Postnatal cortical neurons were isolated following a protocol adapted from Brewer and Torricelli (2007) and the Miltenyi Biotec MACS protocol for the isolation and cultivation of neurons from adult mouse brain. Reagents (Table S11) and contents of MACS Adult Brain Dissociation Kit (130-107-677; Miltenyi Biotec) and Adult Neuron Isolation Kit (130-126-602; Miltenyi Biotec) were used. Cortices of 4-wk-old mice were dissected and transferred to dishes containing prechilled PBS with calcium, magnesium, glucose, and pyruvate (14287080; Gibco). Cortices were cut into quarters and transferred to gentleMACS C tubes (130-093-237; Miltenyi Biotec) containing 2 ml HAG medium with 2 mg/ml Papain (LS003119; Worthington) and placed on ice. Once all samples were collected, each was placed on the gentleMACS Octo Dissociator (130-096-427; Miltenyi Biotec) with the 37_ABDK_01 program running for 30 min. Once dissociated, samples were passed through a 70-μm MACS SmartStrainer (130-098-462; Miltenyi Biotec) prewetted with PBS and sitting over a 15-ml Falcon tube. The strainer was washed with 2 ml PBS and the volume in the tube adjusted to 3.1 ml. 900 μl of debris removal solution was added and mixed by inverting. A further 4 ml was slowly and carefully pipetted down the side of the tube to layer over the dissociated brain solution with minimal mixing of the layers. Samples were centrifuged at 3,000*g* for 10 min at 4°C to create a gradient, followed by removal of the top clear layer and middle debris-containing layer (~4–5 ml). 8 ml of PBS was added to the remaining solution and centrifuged at 1,000*g* for 10 min at 4°C. The supernatant was removed, and the pellet was resuspended in

1 ml of 1x Red Blood Cell Removal Solution (130-107-677; Miltenyi Biotec) and incubated at 4°C for 10 min. 10 ml BSA solution was added to each sample and centrifuged at 300$g$ for 10 min followed by removal of the supernatant. Cells were resuspended in 80 $\mu$l of BSA solution followed by the addition of 15 $\mu$l of non-neuronal cell antibody cocktail (130-126-602; Miltenyi Biotec), then incubated for 5 min at 4°C. 1 ml of BSA solution was added, followed by centrifugation at 300$g$ for 5 min. The supernatant was removed, and the cells were resuspended in 80 $\mu$l BSA solution with 15 $\mu$l anti-biotin beads (130-126-602; Miltenyi Biotec). Samples were mixed and incubated for 10 min at 4°C, and then, the volume was adjusted to 500 $\mu$l with BSA solution. Prechilled LS columns (130-042-401; Miltenyi Biotec) were placed in a MACS magnetic holder and moistened with 3 ml BSA solution. Cell suspensions were then applied to the column, and the flow-through was collected in a new 15-ml tube. The columns were washed with 2 × 1 ml BSA solution, which was also collected, and then, the suspension was centrifuged at 300$g$ for 5 min at 4°C. Cells were resuspended in 100 $\mu$l postnatal cortical neuron culture medium, and live cells were quantified using trypan blue (1450021; Bio-Rad) exclusion for viability with the Luna II automated cell counter (L40001; Logos Biosystems). The desired number of cells was suspended in 300 $\mu$l of culture medium plated in each well of Nunc Lab-Tek II 8-well chamber slides (C7057; Merck) precoated with 100 $\mu$g/ml poly-D-lysine (P6407; Sigma-Aldrich).

To measure neurite growth, 45,000 cells per well were plated to ensure outgrowth at sufficiently low density to allow quantification. After 5 d of culture, cells were stained for beta-III tubulin, then counterstained with 0.1 $\mu$M DAPI (D9542; Merck) to identify nuclei. Slides were imaged using an Axioplan 2 Zeiss fluorescent microscope and camera. Images were analysed semi-automatically with FIJI software and AutoNeuriteJ plugin to measure neurite length.

## Flow cytometry

Intranuclear fluorescence staining and flow cytometry were used to determine histone acetylation levels in peripheral white blood cells. 200 $\mu$l of peripheral blood was collected from mice in EDTA tubes. Red blood cells were depleted using a hypotonic red blood cell lysis buffer (Table S11). Dead cells were stained using a LIVE/DEAD Fixable stain (L23101; Thermo Fisher Scientific) in PBS for 30 min on ice. Cells were washed in PBS and centrifuged (200$g$, 5 min). Cells were resuspended in 2% FACS buffer (Table S11) containing cell surface antibodies (Table S12) for 1 h on ice. Cells were washed in 2% FACS buffer and centrifuged (200$g$, 5 min) before being fixed and permeabilised using the FOXP3 transcription factor kit (00-5523-00; Thermo Fisher Scientific) for 1 h on ice. Cells were washed using 2% FACS buffer for 1 h to remove the fixation buffer. Cells were resuspended in 2% FACS buffer containing a primary histone acetylation antibody (Table S12) and incubated O/N at 4°C on a roller. The next morning, the cells were washed using 2% FACS buffer and incubated in secondary antibody (Table S12) for 1 h on ice. Samples were analysed using a flow cytometry analyser (FACS LSR II SORP or Fortessa X20, BD Biosciences) at a rate of less than 7,500 events per second.

## Acid protein extraction to recover histone proteins from mouse tissues

Frozen spleen samples of mice were crushed with a frozen steel mortar and pestle over dry ice, then transferred to a 1.5-ml Eppendorf tube with a spatula. The powder was suspended in 1 ml nuclear isolation buffer (NIB; Table S11) with 0.2% NP-40 alternative buffer. The suspension was incubated on ice for 30 min with occasional gentle pipetting, then centrifuged at 2,500$g$ for 10 min at 4°C. The supernatant was removed, and the nucleus pellet was washed three times in NIB at a 1:10 (vol/vol) ratio. Nuclei were resuspended in 0.2 M H$_2$SO$_4$ (added at 1:5 vol/vol) and incubated at 4°C for 4 h with constant rotation. Suspensions were centrifuged at 5,000$g$ for 5 min at 4°C, supernatants were transferred to a new tube, and TCA buffer was added to a final concentration of 33% and incubated at 4°C overnight to precipitate histone proteins. The next day, the suspensions were centrifuged at 8,500$g$ for 5 min at 4°C, and the supernatants were removed. The precipitated protein was rinsed with 1 ml ice-cold acetone with 0.1% HCl, and centrifuged at 8,500$g$ for 5 min at 4°C. Two further washes were conducted with 1 ml ice-cold acetone. The pellets were dried, and the protein was dissolved in 100 $\mu$l of ddH$_2$O, and centrifuged at 8,500$g$ for 5 min, and supernatants were transferred to a new tube. Protein concentration was measured with Pierce BCA Protein Assay Kit (23225; Thermo Fisher Scientific) according to the manufacturer's instructions, after which samples were stored at –80°C until required.

Frozen brain hemisphere samples were crushed with a frozen steel mortar and pestle over dry ice, then transferred to a 2-ml Eppendorf tube with a spatula. The powder was suspended in 2 ml of histone lysis buffer (10 mM Hepes [pH 7.9], 1.5 mM MgCl$_2$, 10 mM KCl in Milli-Q H$_2$O with the addition of 0.6 mM DTT, sodium butyrate, and 1x cOmplete protease inhibitor EDTA-free immediately before use) and incubated on a roller for 30 min at 4°C. The suspension was pelleted through centrifugation at 11,000$g$ for 10 min at 4°C, and the supernatant was discarded. The pellet was resuspended in 800 $\mu$l 0.2 M sulphuric acid and incubated on a roller for 2 h at 4°C, followed by centrifugation for 10 min at 11,000$g$ to remove cellular debris. The supernatant was transferred to dialysis membrane regenerated cellulose tubing with the pore size of 3.5–5 kD (133192; Spectra/Por) and dialysed against 0.1 M acetic acid for 1 h, then against Milli-Q H$_2$O overnight. Extracted proteins were removed from the dialysis casing into 1.5-ml Eppendorf tubes, and concentration was measured with Pierce BCA Protein Assay Kit (23225; Thermo Fisher Scientific) according to the manufacturer's instructions, after which samples were stored at –80°C until required.

## Western immunoblotting

Protein samples were denatured with 2 x SDS buffer, separated on NuPAGE 4–12% Bis-Tris Gel (NP0335; Invitrogen), and blotted onto nitrocellulose membranes (926-31092; LICORbio). Membranes were blocked with intercept buffer (927-70001; LICORbio) for 1 h, then incubated with primary antibody overnight at 4°C. Different membranes for each primary antibody were used, each then

incubated with a pan-H3 loading control antibody and secondary antibodies with intervening three 5-min washes with phosphate buffer saline with 0.1% Tween-20 (0.1% PBS-T). After final three 5-min washes in each of 0.1% PBS-T and PBS, the blots were imaged on Odyssey CLx (LICORbio) with Image Studio Lite software. Densitometry was performed using Image Studio Software (LICORbio).

## Cresyl violet staining of adult brains

Adult mouse brains were fixed in Bouin's fixative or neutrally buffered formalin solution. Brains were dissected and paraffin-embedded, and 7-µm coronal sections were stained with cresyl violet. Volumetric analysis was carried out as described previously (West et al, 1991; Coggeshall, 1992).

## RNA sequencing

To analyse the role of KAT6A in gene regulation in the developing brain, E12.5 dorsal telencephalon tissue was dissected and frozen, and E16.5 cortical neurons were isolated and total RNA was extracted using an RNeasy mini kit (QIAGEN) according to the manufacturer's instructions including the optional DNase I digestion step. RNA concentration and quality were determined with an Agilent 2200 TapeStation (Agilent). Stranded mRNA libraries were generated with a TruSeq Stranded mRNA (100 ng plus) kit (Illumina) according to the manufacturer's instructions. The samples were pooled and sequenced with a NextSeq 500 sequencing machine (Illumina) to give 61-bp paired-end reads.

## RNA-sequencing data analysis

Samples were aligned to a combined reference genome comprising *Mus musculus* (mm10) and *Drosophila melanogaster* (release R6.35) using Rsubread (v2.2.6) (Liao et al, 2019). In all samples, the proportion of fragments successfully mapped to the genome exceeded 91%. Gene-level counts were obtained using featureCounts with GENCODE (vM25) annotation for mouse and *D. melanogaster* annotation from FlyBase (R6.35).

The E12.5 and E16.5 datasets were analysed separately. Lowly expressed genes were filtered out using edgeR's filterByExpr function (Chen et al, 2025). Library sizes were normalised using the trimmed mean of M-values (TMM) method (Robinson & Oshlack, 2010). Differential expression (DE) analysis was performed using the limma (v3.44.3) package (Ritchie et al, 2015) with the limma-trend pipeline (Law et al, 2014) and robust empirical Bayes estimation of variances (Phipson et al, 2016). False discovery rate (FDR) control was applied using the Benjamini–Hochberg method at a threshold of 0.1.

For the E12.5 data, sample-specific relative quality weights were estimated using arrayWeights (Ritchie et al, 2006), and three surrogate variables computed by limma's wsva function were included to account for hidden confounders. For the E16.5 data, adjustment for mouse litter was included in the model design.

## Mouse behavioural testing

Behavioural testing was conducted as described previously (Bergamasco et al, 2024). The tests were completed during the light cycle. Mice moved through the tests in cohorts of 8–13 in approximately equal numbers for each genotype and sex. To record movement for later analysis, a camera was placed overhead in all behavioural tests apart from motor function and strength tests, where a front view camera was required. Tests were analysed with TopScan LITE software (CleverSys) where possible. Motor function and strength tests, spontaneous alternation in the Y maze, sociability, Barnes maze trials, and visual cliff tests were analysed manually.

### Habituation to the operator and the testing environment

At 5–6 wk of age in the week before behavioural testing, mice were habituated to the experimenter over three consecutive days through a gradual increase in handling in order to reduce handling-related anxiety (Deacon, 2006). All tests were performed by the same experimenter that mice were habituated to. Before each test, mice were moved into the behavioural procedure room for 30 min to acclimatise to the environment. All equipment was wiped with 80% ethanol before and between each test mouse to remove olfactory cues.

### Developmental milestones

Developmental milestones (Heyser, 2004) were assessed from postnatal days 1 to 21.

### Ultrasonic vocalisation

Individual pups were separated from their mother and placed into a sound-attenuated chamber, prewarmed to nesting temperature (~34°C). Mice were separated and recorded for 3 min on postnatal days 4, 8, and 12. Adult male mice were exposed to females in oestrus for 3 min, and their vocalisations were recorded for 3 min. Vocalisations were visualised and analysed using bioacoustics software (Avisoft-SASLab Pro).

### Home cage analysis

An automated home cage analysis was performed to assess the general activity and behaviour of the mice without interference of an operator using an automated home cage analyser equipped with microchip detectors at the bottom and a camera (Actual Analytics). Microchips were inserted subcutaneously in the lower abdominal region under general anaesthesia 1 wk before testing. Cohorts of 2–4 mice that had been previously housed together were analysed over a 24-h period (Bains et al, 2016).

### Anxiety and locomotion

**Large open field** The large open field test (Heyser, 2004; Samson et al, 2015) was used to assess locomotion and anxiety-like behaviour in an exposed environment. The field was circular with a diameter of 90 cm and ceiling height of 1.5 m and was fully enclosed in a curtain to remove external visual cues and control light intensity (90 lux). Mice avoid exposed open areas, preferring to explore the periphery of the field close in proximity to the maze walls. More exploration towards the centre of the field was therefore indicative of decreased anxiety-like behaviour. Each mouse was placed in the centre of the open field, and movement was recorded for 20 min for the baseline assessment of behaviour, and 5 min when assessing the effect of ALCAR treatment. The

distance that each mouse travelled and the proportion of time spent exploring the central 45 cm of the field were determined.

**Elevated zero maze** The elevated zero maze (EZM [Tucker & McCabe, 2017]) was used to assess anxiety-like behaviour in an open and elevated environment. The experiment was conducted in a 40-cm elevated, 5-cm-wide circular course comprised of alternating closed and open quadrants. The closed quadrants were enclosed by 20-cm-high walls, and the open quadrants only had a 0.5-cm barrier while being exposed to the room. Light intensity recorded in the open areas was 70 lux. Each mouse was placed beside a closed boundary, then allowed to explore for 5 min while being recorded. The proportion of time spent in open areas, the number of open-section entries (where the entire body has crossed into the open area), and the distance that each mouse had travelled were determined.

**Light/dark box** The light/dark box was used to assess anxiety-like behaviour in a bright environment. The apparatus comprised of two halves of 20 × 40 cm zones, with the dark zone enclosed with black walls and ceiling (light intensity 5 lux) with a small doorway leading to the light zone. The light zone had three white walls with an open roof with a bright fluorescent light illuminating the zone evenly (light intensity 330 lux). Each mouse was placed adjacent to the doorway and allowed to explore for 10 min. During this time movement in the light zone was recorded from above. The proportion of time spent in the light zone and number of dark/light transitions were determined.

### Learning and memory

The Y maze (Kraeuter et al, 2018) was used to assess spatial working memory and spatial recognition memory.

**Y maze—spatial working memory** The spontaneous alternation test was used to assess spatial orientation and working memory. It was performed on a Y maze, comprised of three arms of equal size (38 cm in length) enclosed by 12-cm-high walls. Mice relied on external visual cues around the room for spatial orientation. Mice were placed in the centre of the maze and allowed to explore for 5 min. Spontaneous alternation was considered a trio sequence of entries into three different arms (e.g., A-B-C, B-C-A) as mice with strong working memories are more likely to enter an arm least visited rather than one visited directly prior. Entries were manually counted as whole body entries into an arm, and the percentage of alternation was calculated as the number of trios divided by total entries minus 2, as the last two entries did not mark the beginning of a trio.

**Y maze—spatial recognition memory** The novel arm Y maze test was used to assess short-term spatial recognition memory and exploration of novelty. It was performed on the same Y maze as spontaneous alternation, however, with distinguishing internal visual cues above the ends of each arm to aid in orientation. The test was performed at least a week after spontaneous alternation for mice that underwent both tests. Each mouse commenced the experiment in one arm, designated the home arm, and was allowed to explore for 10 min. In this phase I of the test, one arm was blocked, allowing access to only the home arm and one other arm. 1 h later, each mouse underwent a further 5 min of exploration. In this phase II of the test, the home arm, the arm explored in phase I, designated the familiar arm, and the third arm previously closed arm, designated the novel arm, were open for exploration. The proportion of time spent and distance travelled in the novel and familiar arms, as well as the number of arm entries, were determined.

**Novel object recognition test** The novel object recognition (NOR) test (Ennaceur & Delacour, 1988; Bevins & Besheer, 2006; Leger et al, 2013; Samson et al, 2015) was used to assess long-term recognition memory and exploration of novelty. The arena used was the open field, with mice habituated to the empty area for 5 min the day before the NOR test. In the test, two identical Lego towers (70 mm high, 63 mm wide, 63 mm deep) were placed opposite each other; the test mouse was placed in the centre and allowed to explore the objects for 10 min. After a 5-h interval, one of the now familiar objects was replaced with a novel object similar in height with a distinctly different shape. Each mouse was then returned to the centre of the arena for a further 10 min of exploration. The mice were scored for their interaction with each object, and detected automatically in TopScan LITE as the tip of the nose entering within the boundary area of the object, ~2 cm from the object's surface. The proportion of time spent exploring the novel and familiar objects, as well as total exploration times, was determined.

**Barnes maze** The Barnes maze (Bach et al, 1995; Mayford et al, 1996) was used to assess spatial learning and memory through training a mouse to seek and enter a target hole. The operator was located behind a curtain. The Barnes maze (MazeEngineers) was comprised of a white circular platform 92 cm in diameter and 95 cm high with 20 circular holes (5 cm diameter) around the periphery equal distances apart. Underneath was a black "false floor" covering the base of 19 holes, with a black escape tunnel, step leading to a dark escape chamber under the final hole, designated the target hole. Overhead were two LED lights delivering a light intensity of 360 lux for the purpose of creating an adverse environment to motivate seeking of the target hole. A large visual cue on each wall aided in spatial orientation. Mice were habituated to dropping into the target hole the day before the first training session. Mice were placed in the centre of the maze underneath a cover for ~10 s, before being released to seek the target hole. The time taken to (a) find and (b) enter the target hole was recorded, as well as the number of holes visited before the target was determined. If the mice did not enter the target hole within 3 min, they were directed to the target and encouraged to enter. Each mouse underwent four trials a day, with 15-min intervals between trials, across 4 d, and the average of the four trials per day was recorded manually. TopScan LITE software (CleverSys) was used to determine the amount of time spent at each hole, as well as the number of hole visits.

### Sociability

**Three-chamber social recognition** The three-chamber social recognition test (Kaidanovich-Beilin et al, 2011) was used to assess

sociability. The apparatus comprised three chambers, two outer chambers of equal size, and a narrower centre chamber with a grid floor. The external walls were transparent, and the internal walls dividing the chambers were black, with a 3-cm gap underneath to allow the test mouse to move between chambers. All stranger mice had been habituated to a wire cage and apparatus over the previous 3 d, and a beaker with weights was placed on top of the wire cage to prevent extensive movement of the stranger mouse, as well as to prevent the test mouse climbing on top. First, each test mouse was habituated to the empty chambers for 5 min, 1 h before the sociability test. In the sociability test, an empty wire cage was placed in one chamber, whereas a wire cage containing a con-specific stranger mouse was placed in the opposite chamber. The test mouse explored the chambers for 10 min. The proportion of time exploring the empty cage and stranger mouse was analysed manually. Exploration was considered as the tip of the test mouse nose being within 2 cm of the wire cage, either sniffing or in direct contact, as well as rearing against the cage.

### Discrimination index

The preference for and therefore recognition of the novel object, the novel arm in the Y maze, or the mouse over the empty cage were assessed by calculating the discrimination index (Ennaceur & Delacour, 1988):

$$\text{Discrimination Index} = \frac{(\text{Time exploring the novel object} - \text{Time exploring the familiar object})}{(\text{Time exploring the novel object} + \text{Time exploring the familiar object})}$$

The discrimination indices of all genotype and treatment groups were compared individually with a theoretical value of 0, which would indicate no preference for the novel or familiar object, Y maze arm, or cage with mouse over empty cage using a one-sample $t$ test. Between genotypes, discrimination indices were compared by an unpaired, two-tailed $t$ test.

### *Motor coordination and strength*
**Grip strength assessment** Grip strength was tested using a grip strength meter (BIO–GS3; Bioseb). Mice were held by the tail, lowered towards the meter until they grabbed the bar with both forepaws, and gently pulled away from the bar. The maximum force used by the mouse before releasing the bar was recorded. Mice were tested in five trials, separated by a 1-min rest. The best three of these trials were used to compare grip strength between genotypes.

**Rotarod** The rotarod (Arbogast et al, 2015; Luh et al, 2017) was used to assess motor coordination. The rod was ~3 cm in diameter with a ridged surface supported 15 cm above the base, where padding was provided to cushion falls. The speed of the rod rotation could be adjusted manually. In every test, a heterozygous and WT mouse was tested at the same time in different compartments. The day before conducting the main experiment, mice underwent three consecutive trials (with a 1-min intertrial interval) of 2 min at a speed of 12 rpm to habituate to the lowest functioning speed on the apparatus. On the test day, 7 consecutive trials of 2 min per trial (with 1-min intertrial intervals) were undertaken with the following set speeds: 12, 16,

20, 25, 30, 35, and 40 rpm. This series was repeated after 1 h, and the results were averaged to calculate the average time on the rotarod.

**Adhesive removal** The adhesive removal test (Bouet et al, 2009) was used to assess sensory function and fine motor coordination. In this test, each mouse was habituated to the test cage for 1 min before being held by the scruff. A small piece of adhesive tape was attached to each palm of the forepaw, and the mouse was placed in the test cage. Video monitoring was used to record time to begin removing the tape, total time removing tape from first contact, and time spent actively removing tape from paws as many mice paused between removing the first and second pieces. The test was performed on each mouse once a day over 5 d, and the results were averaged.

### *Visual acuity*
**Visual cliff** The visual cliff test was used to assess visual function by creating a perceived barrier and a false cliff. The apparatus was a 62 × 62 cm transparent acrylic sheet chamber, half of which overhung from a support table, 50 cm above the floor. A red-and-white checker pattern was placed directly underneath the portion of the chamber supported by the table, as well as on the floor below the overhanging portion to create an illusion of a cliff and aid in-depth perception. A camera was directly overhead to record movement. 4–24 h before the commencement of the task, each mouse had their whiskers trimmed to help minimise sensory detection of the transparent floor. Each mouse was placed in the shallow side of the visual cliff and allowed to explore the chamber for 5 min before returning to the home cage. Latency to cross over the cliff, number of crossings, and time spent on the shallow side, over the cliff, and in the boundary zones were analysed manually.

### Statistical analysis

Data were analysed using the GraphPad Prism version 7 (GraphPad Software) with the exception of the analysis of RNA-sequencing data, which is described in the Materials and Methods section under RNA-sequencing data analysis. The number of observations and statistical tests used are stated in the figure legends.

## Data Availability

RNA-sequencing data are available at the NCBI Gene Expression Omnibus (GEO accession number: GSE304794). Gene expression levels and fold changes of all genes detected are available in Tables S4, S5, and S6.

## Supplementary Information

# Life Science Alliance

# Acknowledgements

The authors thank L Johnson, L Wilkins, S Bound, S Oliver, J Martin, R Meeny, N Blasch, and F Dabrowski for animal care, S Holloway and J House for veterinary care, L Potenza and C Burström for excellent technical assistance, and A Samson for insightful advice. HK Vanyai was supported by the AI & Val Rosenstrauss Fellowship from the Rebecca L. Cooper Medical Research Foundation. This work was supported by the Lorenzo and Pamela Galli Medical Research Trust; the Valda Klaric Foundation; the Australian National Health and Medical Research Council through project grant 1160517 to T Thomas, Ideas Grant 2010711 to T Thomas, Research Fellowships 1081421 to AK Voss and 1154970 to GK Smyth, and Investigator Grants 1176789 to AK Voss and 1194345 to ME Blewitt; through the Independent Research Institutes Infrastructure Support Scheme; and by the Victorian Government through an Operational Infrastructure Support Grant.

## Author Contributions

S Eccles: formal analysis, investigation, project administration, and writing—review and editing.
HK Vanyai: formal analysis, investigation, and writing—review and editing.
MI Bergamasco: formal analysis, investigation, and writing—review and editing.
S Malelang: investigation.
H Pehlivanoglu: investigation.
AL Garnham: formal analysis, investigation, and writing—review and editing.
N Ranathunga: formal analysis, investigation, and writing—review and editing.
ME Blewitt: supervision and writing—review and editing.
AP Vogel: methodology and writing—review and editing.
GK Smyth: formal analysis, supervision, methodology, and writing—review and editing.
AJ Hannan: methodology and writing—review and editing.
T Thomas: conceptualisation, funding acquisition, and writing—review and editing.
AK Voss: conceptualisation, resources, formal analysis, supervision, funding acquisition, investigation, visualisation, methodology, project administration, and writing—original draft, review, and editing.

## Conflict of Interest Statement

AK Voss and T Thomas are inventors on patent WO2016198507A1. AK Voss and T Thomas have received research funding from the Cancer Therapeutics CRC (CTX). AK Voss and T Thomas have served on a clinical advisory board for Pfizer.

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
