## [Reviewer comments · Life Science Alliance]

Acetyl-carnitine improves hyperactivity and learning deficits in KAT6A haploinsufficient mice

Samantha Eccles, Hannah Vanyai, Maria Bergamasco, Shezlie Malelang, Havva Pehlivanoglu, Alexandra Garnham, Nishika Ranathunga, Marnie Blewitt, Adam Vogel, Gordon Smyth, Anthony Hannan, Tim Thomas, and Anne Voss

DOI: <https://doi.org/10.26508/lsa.202503549>

Corresponding author(s): Anne Voss, Walter and Eliza Hall Institute of Medical Research and Tim Thomas, Walter and Eliza Hall Institute

Review Timeline:

Submission Date:	2025-10-25
Editorial Decision:	2025-11-26
Revision Received:	2026-01-17
Editorial Decision:	2026-01-28
Revision Received:	2026-02-03
Accepted:	2026-02-08

Scientific Editor: Tim Fessenden

Transaction Report:

November 26, 2025

Re: Life Science Alliance manuscript #LSA-2025-03549-T

Prof. Anne K. Voss
Walter and Eliza Hall Institute of Medical Research
Development and Cancer Division
1G Royal Parade
Melbourne, Parkville, Victoria 3052
Australia

Dear Dr. Voss,

Thank you for submitting your manuscript entitled "ALCAR treatment improves hyperactivity and learning deficits in KAT6A haploinsufficient mice" to Life Science Alliance. The manuscript was assessed by expert reviewers, whose comments are appended to this letter.

As you will see, both reviewers found this work interesting and potentially valuable for this field. However, both noted claims that should be adjusted in light of the data shown. In particular, Reviewer 2 expressed concern over the discordant results shown in Fig 5 vs Fig 7 on novel object tests. We appreciate the attention of Reviewer 2 to this important issue although we do not favor removing these observations from the manuscript. A revised manuscript should address this issue in greater depth in the results and discussion sections. In line with this limitation, and in accordance with Reviewer 1, a revised manuscript must restate the therapeutic potential of ALCAR treatment more carefully in the abstract. Finally, we concur with Reviewer 2 in point 2 that this work would be strengthened by a discussion of any relevant results from Liu et al on changes in gene expression in adult brains to connect these with adult behaviors. Points 3 and 4 by Reviewer 2 can be addressed with changes to the text as needed.

I would be happy to discuss the revision in more detail via email or phone/videoconferencing. Please let me know which option you prefer, if any.

While you are revising your manuscript, please also attend to the below editorial points to help expedite the publication of your manuscript. Please direct any editorial questions to the journal office. When submitting the revision, please include a letter addressing the reviewers' comments point by point.

Thank you for this interesting contribution to Life Science Alliance. We hope that the comments below will prove constructive as your work progresses, and we are looking forward to receiving your revised manuscript.

Sincerely,

B. MANUSCRIPT ORGANIZATION AND FORMATTING:

Reviewer #1 (Comments to the Authors (Required)):

This study focuses on acetyl-carnitine treatment in mouse models for KAT6A syndrome. The results are important and timely. The only concern that I have is that the authors need to be cautious that related patients may use the results to take acetyl-carnitine, as many patients and their families are desperate for something to deal with the disease. Thus, it is important to note this caution in the manuscript, perhaps in the abstract.

One minor comment is the dosage 100 mg/kg. How it was chosen should be detailed in the manuscript. I would also suggest to use acetyl-carnitine, instead of the acronym in the title.

Reviewer #2 (Comments to the Authors (Required)):

Summary

Eccles et al. characterize KAT6A haploinsufficiency on a molecular and behavioral scale and explore acetyl-L-carnitine (ALCAR) as a therapeutic correction for Arboleda-Tham syndrome (ARTHS). Authors show that KAT6A mutations lead to decreased H3K23 acetylation in human cells and mouse brain. The *Kat6a*^{+/-} mice are hyperactive with learning/memory deficits and decreased sociability. When treated, ALCAR treatment increases H3K23ac levels and restores hyperactivity and learning deficits but not sociability deficits. The study is comprehensive and clinically translationally relevant. The progression of study from human ARTHS mutations to cell line modeling to mouse modeling to potential therapeutic approaches make this study relevant and add great clinical relevance. The extensive behavioral battery used is an extensive and comprehensive characterization of the observed phenotype and showing that ALCAR mediates effects by specifically rescuing H3K23ac levels provides an effective mechanistic rationale for findings. However, several major concerns exist that must be addressed.

Comments

1) The major issue of the study is the missing comparison of baseline vs. treatment cohorts as it pertains to positive findings from ALCAR efficacy. For example, Figure 5B-J shows that the *Kat6a*^{+/-} mice have decreased preference for the novel arm/object in the Y-maze and novel object recognition, but, in the vehicle-treated cohort during ALCAR testing, these differences largely disappear (Figure 7G-K). In addition, the authors mention that "saline injections and mash were supportive treatment" but this is highly concerning. If the vehicle treatment itself is therapeutic, then it becomes impossible to determine ALCAR's specific effects in these tests. Without showing that vehicle treated mice recapitulate the baseline phenotype, the conclusions about ALCAR efficacy in spatial and object recognition memory are not supported. The authors state this "hypothesis awaits formal testing" but this testing should have been done to support claims made here. Therefore, I strongly suggest that either: a) this experiment be appropriately conducted with a vehicle control that recapitulates the baseline deficits or b) these findings be supported with additional experiments showing the vehicle effect is real, reproducible, with mechanistic rationale or c) removing these tests from the treatment section and focusing conclusions only on phenotypes that were consistently observed across both baseline and vehicle conditions (hyperactivity in open field, Barnes maze spatial learning).

2) Another concern is regarding the time of molecular and behavioral analysis. For example, while RNA-seq data is assessed in E12.5 and E16.5 embryonic tissue, behavioral analysis is assessed in 6-8 week old adults. While the authors mentioned the gene expression changes are reported in adult brain by Liu et al. (2024), they do not integrate these findings. The manuscript would benefit hugely from assessing RNA-seq data of adult brain regions of interest; or providing more detailed analysis and

discussion of how developmental gene expression changes might mechanistically lead to adult behavioral phenotypes. The current presentation leaves a significant gap in understanding the progression from embryonic molecular changes to adult functional deficits.

3) The mechanistic understanding of how ALCAR achieves its effects is incomplete. While the authors show that ALCAR treatment restores H3K23ac levels, there are many remaining questions which should be addressed: How does systemic ALCAR treatment increase brain H3K23ac? Does ALCAR increase acetyl CoA levels in brain? Is there global effect for other histone marks with regard to treatment? Finally, the selectivity of ALCAR impact on certain aspects compared to others ARTHS components should be acknowledged: Why does ALCAR rescue hyperactivity and learning deficits but not sociability? The authors should measure acetyl-CoA levels in brain tissue to confirm the presumed mechanism and should provide more detailed discussion of the selective rescue of certain phenotypes. This selective effect could provide important insights into the neurobiological basis of different ARTHS symptoms but is currently underexplored.

4) I think the statement "brain development continues after birth" is a bit vague. Authors should specify which aspects of development continue, if any of them are regionally applicable/what time frames in relation to when treatment begins. This is crucial for treatment rationale, and clinical translation. Additionally, there is no mention of what happens if ALCAR is stopped. Are the beneficial effects sustained, or are they reversible? This has important implications for understanding whether ALCAR is correcting developmental deficits or providing ongoing support.

Manuscript #LSA-2025-03549-T

We would like to thank the editor and the reviewers for the time and effort spent on reviewing our manuscript. We believe that the changes made in response to the comments have improved our manuscript.

Point-by-point responses to the Reviewers' comments**Reviewer #1** (Comments to the Authors (Required)):

Reviewer #1: This study focuses on acetyl-carnitine treatment in mouse models for KAT6A syndrome. The results are important and timely.

Response: We thank the reviewer for the positive comments on importance and timeliness of our work.

Response: The only concern that I have is that the authors need to be cautious that related patients may use the results to take acetyl-carnitine, as many patients and their families are desperate for something to deal with the disease. Thus, it is important to note this caution in the manuscript, perhaps in the abstract.

Response: We agree and thank the reviewer for this comment. As suggested, we have included a cautionary statement at the end of the abstract.

Reviewer #1: One minor comment is the dosage 100 mg/kg. How it was chosen should be detailed in the manuscript.

Response: As requested, we have now explained the choice of the ALCAR dosage in the methods section on page 40.

Reviewer #1: I would also suggest to use acetyl-carnitine, instead of the acronym in the title.

Response: As the reviewer suggested, we have now replaced ALCAR with acetyl-carnitine in the title.

Reviewer #2 (Comments to the Authors (Required)):**Reviewer #2:** Summary

Eccles et al. characterize KAT6A haploinsufficiency on a molecular and behavioral scale and explore acetyl-L-carnitine (ALCAR) as a therapeutic correction for Arboleda-Tham syndrome (ARTHS). Authors show that KAT6A mutations lead to decreased H3K23 acetylation in human cells and mouse brain. The *Kat6a*^{+/-} mice are hyperactive with learning/memory deficits and decreased sociability. When treated, ALCAR treatment increases H3K23ac levels and restores hyperactivity and learning deficits but not sociability deficits. The study is comprehensive and clinically translationally relevant. The progression of study from human ARTHS mutations to cell line modeling to mouse modeling to potential

therapeutic approaches make this study relevant and add great clinical relevance. The extensive behavioral battery used is an extensive and comprehensive characterization of the observed phenotype and showing that ALCAR mediates effects by specifically rescuing H3K23ac levels provides an effective mechanistic rationale for findings. However, several major concerns exist that must be addressed.

Response: We thank the reviewer for positive comments on the clinical relevance and completeness of our work and for noting the extensive behavioural work that has gone into this paper.

Reviewer #2: Comments

1) The major issue of the study is the missing comparison of baseline vs. treatment cohorts as it pertains to positive findings from ALCAR efficacy. For example, Figure 5B-J shows that the *Kat6a*^{+/-} mice have decreased preference for the novel arm/object in the Y-maze and novel object recognition, but, in the vehicle-treated cohort during ALCAR testing, these differences largely disappear (Figure 7G-K). In addition, the authors mention that "saline injections and mash were supportive treatment" but this is highly concerning. If the vehicle treatment itself is therapeutic, then it becomes impossible to determine ALCAR's specific effects in these tests. Without showing that vehicle treated mice recapitulate the baseline phenotype, the conclusions about ALCAR efficacy in spatial and object recognition memory are not supported. The authors state this "hypothesis awaits formal testing" but this testing should have been done to support claims made here. Therefore, I strongly suggest that either: a) this experiment be appropriately conducted with a vehicle control that recapitulates the baseline deficits or b) these findings be supported with additional experiments showing the vehicle effect is real, reproducible, with mechanistic rationale or c) removing these tests from the treatment section and focusing conclusions only on phenotypes that were consistently observed across both baseline and vehicle conditions (hyperactivity in open field, Barnes maze spatial learning).

Response: We agree with the reviewer that the benefit of ALCAR can only be concluded from the tests where the vehicle treated cohorts replicated the baseline analysis, i.e., the hyperactivity and the spatial learning and memory in the Barnes maze. That was our intention and that is why we restricted our conclusions to the hyperactivity and spatial learning and memory in the Barnes maze. To address the reviewer's comment, we have now added new data supporting our hypothesis that fluid or nutritional supportive treatment in the postnatal period when *Kat6a*^{+/-} mice fail to thrive may improve their development. The new data were added to Figure S8 as panels S8N and S8O.

The discussion of this phenomenon (in the original manuscript on pages 21 and 22) has been amended in view of the new results (now page 24).

Reviewer #2: 2) Another concern is regarding the time of molecular and behavioral analysis. For example, while RNA-seq data is assessed in E12.5 and E16.5 embryonic tissue, behavioral analysis is assessed in 6-8 week old adults. While the authors mentioned the gene expression changes are reported in adult brain by Liu et al. (2024), they do not integrate these findings. The manuscript would benefit hugely from assessing RNA-seq data of adult brain regions of interest; or providing more detailed analysis and discussion of how developmental gene expression changes might mechanistically lead to adult behavioral phenotypes. The current presentation leaves a significant gap in understanding the progression from

embryonic molecular changes to adult functional deficits.

Response: When deciding what tissues to use for our RNA sequencing analysis we considered the following:

(1) ARTHS presents with global developmental delay from birth indicating strongly an involvement of brain development.

(2) The comparison relevant to ARTHS individuals, who have pathogenic variants in just one allele of the *KAT6A* gene would be *Kat6a*^{+/-} vs. wild type mice. In our experience that poses a problem because the loss of just allele of any gene typically causes only subtle differences in gene expression.

(3) RNA sequencing experiments are typically sensitive and successful if the starting material is as homogeneous as possible, ideally only one cell type. We chose E12.5 dorsal telencephalon and E16.5 cortical neurons for the following reasons:

Regarding point (1): Material from embryonic day 12.5 and 16.5 addresses potential effects of loss of one allele of *Kat6a* on brain development.

Regarding point (2): *Kat6a*^{-/-} embryos on a C57BL/6 genetic background die between E13.5 and E14.5. At E12.5 we are able to isolate homozygous null, as well as heterozygous and wild type embryos. This allowed us to examine gene expression differences between *Kat6a*^{-/-} and *Kat6a*^{+/+} samples and compare these to gene expression differences between *Kat6a*^{+/-} and *Kat6a*^{+/+} samples. This approach affords a method to address the problem of the subtle gene expression differences typically observed between heterozygous and wild type samples.

Regarding point (3): The dorsal telencephalon of E12.5 embryos on a C57BL/6 genetic background consists to >90% of proliferating neuroepithelial cells. E16.5 cortical neurons cultured overnight under conditions supporting only neuronal survival consist of approximately 98% neurons.

We believe our focus on the developing brain is justified. We found a large number of gene expression changes (320) in the absence of one allele of *Kat6a* in the dorsal telencephalon, whereas the changes observed later in development and in the adult were more subtle: in our data at E16.5 (20 genes), and in the adult by Liu et al. (2024) (1 gene in adult hippocampus; 90 genes in single nuclei of CA3 pyramidal neurons). We discussed our results in comparison to Liu et al. (2024), on pages 18 and 19 in the original manuscript, now pages 19 and 20 in the revised version of the manuscript.

We did discuss our hypothesis of what might be occurring mechanistically: it was very notable that *Kat6a*^{+/-} vs. wild type E12.5 dorsal telencephalon expressed 282 brain development genes prematurely suggesting premature neuronal differentiation. This implies that the neural precursor pool might be depleted prematurely resulting in too few progenitor cells. We discussed this hypothesis on pages 18 and 19 in the original manuscript, now page 20 in the revised version of the manuscript.

We believe that we compared our results extensively to Liu et al., 2024 in our original manuscript in the locations stated below. We included two paragraphs on Liu et al., 2024 in the discussion and even generated graphs comparing the fold changes in RNA levels between data generated by Liu et al. (2024) and our data. These were displayed in the original Figure S4H and S4I of the manuscript (still Figure S4H and S4I in the revised version). Our data correlated well with the data published by Liu et al. (2024) with R² of 0.55 and 0.67, despite the substantially different sources of RNA, as stated in detail here:

Page 9 in the original manuscript (Page 9 in the revised version): *“Positive correlations were observed between gene expression changes in $Kat6a^{+/-}$ vs. $Kat6a^{+/+}$ dorsal telencephalon and $Kat6a^{+/-}$ vs. $Kat6a^{+/+}$ total adult hippocampus and CA3 pyramidal neurons [GSE261058; GSE261148;(Liu et al, 2024)] despite the diversity of tissue type and age ($R^2 = 0.55$ and 0.67 , respectively; $p = 0.0009$ and 0.002 , respectively; Figure S4H,I).”*

Figure S4H,I (original and revised manuscript supplement):

“(H) Correlation between gene expression changes in $Kat6a^{+/-}$ vs. $Kat6a^{+/+}$ E12.5 dorsal telencephalon ($p \leq 0.003$; $FDR < 0.1$; this study) and $Kat6a^{+/-}$ vs. $Kat6a^{+/+}$ adult hippocampus ($p < 0.05$; GSE261058; (Liu et al., 2024)).

(I) Correlation between gene expression changes in $Kat6a^{+/-}$ vs. $Kat6a^{+/+}$ E12.5 dorsal telencephalon (this study) and $Kat6a^{+/-}$ vs. $Kat6a^{+/+}$ adult CA3 pyramidal neurons ($FDR < 0.008$; GSE261148; (Liu et al., 2024)).”

Page 18 in the original manuscript (Page 19 in the revised version):

“Similar to our findings, Liu and colleagues (Liu et al., 2024) observed that mice lacking exon 3 of the $Kat6a$ gene displayed a deficit in learning and memory manifested as lack of preference for a novel object in the novel object recognition test and an inferior performance in the Morris water maze, which tests similar abilities as the Barnes maze used in our study. Likewise, they found a reduction in natural anxiety as observed in the elevated plus maze, which is similar to our results in the elevated zero maze. A notable difference between the two studies is that we observed hyperactivity and a sociability deficit in $Kat6a^{+/-}$ mice, whereas Liu and co-workers did not. Liu and colleagues did not test treatment options.”

Page 18 in the original manuscript (Page 20 in the revised version):

“Our E12.5 RNA sequencing data revealed a large number of genes upregulated (282) in the absence of one allele of $Kat6a$ in the dorsal telencephalon, the cerebral cortex precursor, whereas more subtle effects in the foetal (our E16.5 data) and adult stage tissue (Liu et al., 2024) were seen. Interestingly, differentially expressed genes in $Kat6a^{+/-}$ vs. $Kat6a^{+/+}$ adult hippocampus (Liu et al., 2024), correlated positively with the top changes in our $Kat6a^{+/-}$ vs. $Kat6a^{+/+}$ E12.5 dorsal telencephalon. The upregulated genes in the $Kat6a^{+/-}$ E12.5 telencephalon were robustly associated with neuron development and maturation, such as axon guidance and synapse organisation. We speculate that this finding may reflect premature neuronal maturation perhaps at the expense of neuronal precursor population expansion which might relate to the cognitive and social deficits.”

In addition, we have now included a new paragraph on Liu et al., 2024 in the discussion on Page 20:

“Liu and co-workers observed impaired synaptic structure and plasticity in the hippocampal CA3 region (Liu et al., 2024). Of 90 genes differentially expressed in the CA3 region, they focussed on $Rspo2$ (r-spondin 2) and show that restoring $Rspo2$ gene expression via adenovirus injection rescues learning-associated deficits in $Kat6a$ mutant mice. It is worth noting that $KAT6A$ is one of only nine nuclear lysine acetyltransferases with a defined acetyltransferase domain, nine proteins that are collectively responsible for all histone

acetylation and other protein acetylation in the nucleus. Accordingly, our data suggest that hundreds of genes depend on KAT6A for their normal expression levels (and histone acetylation). It is therefore likely that Rspo2 is only one of many KAT6A target genes."

We therefore believe that we integrated and discussed the work of Liu et al. (2024) extensively and assessed similarities and difference appropriately. Only those parts of their work that did not have an equivalent in our work were not discussed in detail.

Reviewer #2: 3) The mechanistic understanding of how ALCAR achieves its effects is incomplete. While the authors show that ALCAR treatment restores H3K23ac levels, there are many remaining questions which should be addressed: How does systemic ALCAR treatment increase brain H3K23ac? Does ALCAR increase acetyl CoA levels in brain? Is there global effect for other histone marks with regard to treatment? Finally, the selectivity of ALCAR impact on certain aspects compared to others ARTHS components should be acknowledged: Why does ALCAR rescue hyperactivity and learning deficits but not sociability? The authors should measure acetyl-CoA levels in brain tissue to confirm the presumed mechanism and should provide more detailed discussion of the selective rescue of certain phenotypes. This selective effect could provide important insights into the neurobiological basis of different ARTHS symptoms but is currently underexplored.

Response: We agree with the reviewer, that the predominant mechanism of action of ALCAR responsible for the improvement in hyperactivity and learning performance in *Kat6a*^{+/-} cannot be derived from our data. Likewise, we cannot provide an explanation for the selective improvement in hyperactivity and learning performance, but not in sociability.

We note that we did provide experimental data on other histone acetylation marks in our original paper in Figure S8A to S8H showing that acetylation levels at H3K14 are also elevated by ALCAR treatment in the brain.

To address the reviewer's comment, we have now expanded on how the prevailing understanding of the mechanism by which acetyl-carnitine might promote histone acetylation on page 15 of the revised manuscript. In addition, we have included in the discussion a section on page 22 stating the many known roles of acetyl-carnitine in the brain and have clearly stated that, given the many possible roles, our current data do not identify the predominant role of ALCAR in improving hyperactivity and learning performance, or explain the absence of an effect on sociability.

Reviewer #2: 4) I think the statement "brain development continues after birth" is a bit vague. Authors should specify which aspects of development continue, if any of them are regionally applicable/what time frames in relation to when treatment begins. This is crucial for treatment rationale, and clinical translation. Additionally, there is no mention of what happens if ALCAR is stopped. Are the beneficial effects sustained, or are they reversible? This has important implications for understanding whether ALCAR is correcting developmental deficits or providing ongoing support.

Response: We politely ask the reviewer to kindly note that on page 20 of our original manuscript (page 21 in the revised version of our manuscript), we did elaborate brain developmental processes that continue or occur after birth. There were wrote:

*“Brain development continues after birth into early adulthood (Semple et al., 2013; Stiles & Jernigan, 2010). The brain of a new-born infant is approximately one third of the size of an adult brain (Holland et al, 2014) and a child reaches 95% of the adult brain size at 7 to 11 years of age (Caviness et al, 1996), reflective of substantial growth in the years after birth. While neurons are formed predominantly before birth, glial cells and myelination continue after birth. Pre- and post-natal brain development results in an overabundance (‘developmental exuberance’) of neurons, glia, neurites and synapse connectivity (Innocenti & Price, 2005), which in pre- and post-natal life are consolidated by an activity-dependent mechanism or pruned. The cerebral grey matter reaches its peak at 10 to 12 years of age in the frontal and parietal lobe and at 17 years of age in the temporal lobe and thereafter declines in size, while the white matter increases at least until 22 years of age (Giedd et al, 1999). Post-adolescence changes in the frontal cortex have also been observed (Sowell et al, 1999). The volume of brain regions declines throughout adult life with the exception of the cerebral white matter, which is at its highest levels between 30 and 50 years of age before also declining (Jernigan & Gamst, 2005; Lebel & Beaulieu, 2011). Synapse density peaks at approximately twice the adult levels (for example, between 3.5 and 7 years of age in the prefrontal cortex), before it is reduced by pruning (Huttenlocher & Dabholkar, 1997). Based on this extended timeline of brain development and the continued expression of the *Kat6a* gene in the prenatal to adult brain, it is possible that an effective therapeutic intervention may be beneficial throughout life.”*

As we perceived that our summary of brain development processes that occur after birth came too late in our manuscript for the reviewer, we have now included some brief detail on the processes of brain development that occur after birth in the introduction on page 5, where we now write:

“Brain development continues after birth (Semple et al, 2013; Stiles & Jernigan, 2010) with some processes peaking between after 3 years of age (Huttenlocher & Dabholkar, 1997). For example, mean synaptic density peaks at 8 months of age in the visual cortex and at 3.6 years of age in the auditory and prefrontal cortex, after which synapse elimination eventual results in adult levels of synaptic density during late adolescence (Huttenlocher & Dabholkar, 1997). Grey matter volume peaks at 12 years of age (Semple et al., 2013). Of note, histological and cell biological changes in the developing brain are accompanied by functional changes. For example, executive functions including logical reasoning and judgment appear to develop gradually during childhood and adolescence (Sternberg, 1979; Sternberg & Rifkin, 1979).”

We agree with the reviewer that it would be very interesting to determine if the treatment needs to be continued indefinitely to maintain the improvement in spatial learning and memory and reduction in hyperactivity or if it could be terminated or given intermittently. We are grateful that the reviewer noted the extensive behavioural testing that we conducted. Each new treatment regime would double that work. We therefore put it to the reviewer that alternative treatment regimens should be seen as part of a new study. Our expectation would be that ongoing ALCAR treatment would provide an ongoing improvement, not because there is no developmental aspect to the consequences of loss of one allele of *Kat6a*, but because, as the *Kat6a* gene is expressed in adult brain, it is likely that there are ongoing requirements for KAT6A function.

January 28, 2026

RE: Life Science Alliance Manuscript #LSA-2025-03549-TR

Prof. Anne K. Voss
Walter and Eliza Hall Institute of Medical Research
Development and Cancer Division
1G Royal Parade
Melbourne, Parkville, Victoria 3052
Australia

Dear Dr. Voss,

Thank you for submitting your revised manuscript entitled "Acetyl-carnitine improves hyperactivity and learning deficits in KAT6A haploinsufficient mice". We returned your manuscript to Reviewer 2 who has no further requests. We would be happy to publish your paper in Life Science Alliance pending final revisions necessary to meet our formatting guidelines.

MANUSCRIPT ORGANIZATION AND FORMATTING:

To avoid unnecessary delays in the acceptance and publication of your paper, please read the following information carefully. Full guidelines are available on our Instructions for Authors page, <https://www.life-science-alliance.org/authors>

- Please upload all your Tables in editable .doc or Excel format; They can be included at the bottom of the main manuscript file or be sent as separate files.
- Please upload all figure files individually, including the supplementary figure files; all figure legends should only appear in the main manuscript file.
- Please add your main, supplementary figure, and table legends to the main manuscript text after the references section.
- Please add the second Corresponding Author to our system as well.
- Please add Keywords for your manuscript in our system.
- Please add the X and Bluesky handles of your host institute/organization, as well as your own and/or one of the authors, in our system.
- Please be sure that the authorship listing and order are correct and match between the system and the manuscript file.
- The contributions selected for Marnie Elisabeth Blewitt do not qualify them for authorship. Please either update the contributions in our system and in the Author Contributions section of the manuscript, or let us know if the author needs to be removed (and added potentially to the acknowledgment section).
- Please consult our manuscript preparation guidelines <https://www.life-science-alliance.org/manuscript-prep> and make sure your manuscript sections are in the correct order. LSA requires deposition of RNA seq data to a public repository, which should be provided in a Data Availability statement.
- Please add a callout for Figures 7I; S1A-D; S2D-L; S3A-D; S4B, C, G and S5D-G to your main manuscript text.

Thank you for including source data as uncropped/-processed electrophoretic blots related to the main figures of the manuscript. We encourage you to upload one PDF/Excel-file per figure as Source Data files which will be linked to the main figures in the online publication.

LSA encourages authors to provide a 30-60 second video where the study is briefly explained. We will use these videos on social media to promote the published paper and the presenting author (for examples, see <https://docs.google.com/document/d/1-UWCfbE4pGcDdcgzcmiuJI2XMBJnxKYeqRvLLrLSo8s/edit?usp=sharing>). Corresponding or first-authors are welcome to submit the video. Please submit only one video per manuscript. The video can be emailed to contact@life-science-alliance.org

FINAL FILES:

The following items are required for acceptance.

The license to publish form must be signed before your manuscript can be sent to production. A link to the license to publish form will be available to the corresponding author only. Please take a moment to check your funder requirements.

Thank you for your attention to these final processing requirements. Please revise and format the manuscript and upload materials as soon as you are able.

Thank you for this interesting contribution to the literature. We look forward to publishing your paper in Life Science Alliance.

Sincerely,

Reviewer #2 (Comments to the Authors (Required)):

The authors have addressed my concerns satisfactorily.

February 8, 2026

RE: Life Science Alliance Manuscript #LSA-2025-03549-TRR

Prof. Anne K. Voss
Walter and Eliza Hall Institute of Medical Research
Epigenetics and Development Division
1G Royal Parade
Melbourne, Parkville, Victoria 3052
Australia

Dear Dr. Voss,

Thank you for submitting your Research Article entitled "Acetyl-carnitine improves hyperactivity and learning deficits in KAT6A haploinsufficient mice". It is a pleasure to let you know that your manuscript is now accepted for publication in Life Science Alliance. Congratulations on this interesting work.

DISTRIBUTION OF MATERIALS:

Again, congratulations on a very nice paper. I hope you found the review process to be constructive and are pleased with how the manuscript was handled editorially. We look forward to future exciting submissions from your lab.

Sincerely,
